# On Regularization for Explaining Graph Neural Networks: An Information Theory Perspective

## Abstract

This work studies the explainability of graph neural networks (GNNs), which is important for the credibility of GNNs in practical usage. Existing work mostly follows the two-phase paradigm to interpret a prediction: *feature attribution* and *selection*. However, another important component — regularization, which is crucial to facilitate the above paradigm — has been seldom studied. In this work, we explore the role of regularization in GNNs explainability from the perspective of information theory. Our main findings are: 1) regularization is essentially pursuing the balance between two phases, 2) its optimal coefficient is proportional to the sparsity of explanations, 3) existing methods imply an implicit regularization effect of stochastic mechanism, and 4) its contradictory effects on two phases are responsible for the out-of-distribution (OOD) issue in post-hoc explainability. Based on these findings, we propose two common optimization methods, which can bolster the performance of the current explanation methods via sparsity-adaptive and OOD-resistant regularization schemes. Extensive empirical studies validate our findings and proposed methods. Code is available at https://anonymous.4open.science/r/Rethink_Reg-07F0.

## 1 Introduction

Graph Neural Networks (GNNs) (Dwivedi et al., 2020; Wu et al., 2019) have achieved remarkable progress on various graph-related tasks (Mahmud et al., 2021; Zhao et al., 2021; Guo & Wang, 2021). However, GNNs usually work as a black box, making the decision-making process obscure and hard to interpret (Ribeiro et al., 2016). Hence, answering the question: "What knowledge does the GNN use to make a certain prediction?" is becoming crucial. To solve this question, most prior studies (Yu et al., 2021; Miao et al., 2022) realize post-hoc explainability by extracting the informative yet sparse subgraphs as explanations, following the principle of graph information bottleneck (GIB) (Wu et al., 2020). The common paradigm of these explainers can be summarized as the *relay race* of feature attribution and selection. Specifically, feature attribution distributes the prediction to the input features and traces their importance, and feature selection sequentially fills features into the explanatory subgraph according to the importance rank, where regularization terms are introduced to constrain subgraph properties like size and connectivity.

However, existing explainers allocate little attention to the role of regularization in them, but is the focus of our work. On the one hand, without digging deeper into regularization theoretically, we hardly acquire a plain picture of how regularization specifically affects the process of feature attribution and selection. Furthermore, some regularization in existing explainers lacks concrete theoretical support and is seemingly not more than an empirical trick. For example, GNNExplainer (Ying et al., 2019) leverages the $l_1$ norm to constrain the magnitude of masks and selects the edge with larger importance (*i.e.*, larger mask). The key here is not the absolute magnitude of the mask (i.e., $l_1$ norm), but rather the relative magnitude between the masks. Thus, we argue that the necessity of $l_1$ norm needs more theoretical support.

In sight of this, we endeavor to rethink the role of regularization in GNNs explainability from the perspective of information theory. Before starting, we first reshape the principle of GIB as GIBE (*i.e.*, new GIB form tailored for GNNs Explainability) in the language of feature attribution and

selection. Specifically, GIBE unifies the current explanation methods via formulating the optimization objective of these two phases. It further explores the roles of regularization in two phases respectively. Guided by these explorations, we reveal the essence of regularization and propose four intriguing propositions in terms of it. We believe a better theory of regularization is fundamental:

- **The essence of regularization:** Regularization in GNNs explainability is essentially the tradeoff scheme to pursue the balance between the phases of feature attribution and selection (Section 3.2).
- **On Sparsity:** The optimal coefficients of regularization are proportional to the sparsity of the explanation, that is, high sparsity should require large regularization and vice versa (Section 4.1).
- **On stochastic mechanism:** Existing methods imply an implicit regularization effect of stochastic mechanism, which endows GNNs explainability with better compressibility (Section 4.2).
- **On OOD issue:** The contradictory effects of regularization on two phases are responsible for the OOD issue in the post-hoc explainability (Section 4.3).

Furthermore, based on these findings, we propose two common optimization methods, which can bolster the performance of current explainers via sparsity-adaptive and OOD-resistant regularization schemes. Extensive empirical studies validate our findings and proposed methods in Section 5.

## 2 PRELIMINARY AND RELATED WORK

**GNNs explainability.** While GNNs have achieved remarkable success in node classification (Zhou et al., 2019; Sankar et al., 2019), graph classification (Zhang et al., 2018; Chen et al., 2019), and link prediction (Ying et al., 2018; You et al., 2020) tasks, in this work, we focus on the scenario of interpreting graph classification task comprising the data distribution $\mathcal{D}$ and the classifier $f'$. Specifically, the input graph $\mathcal{G} = (\mathbf{X}, \mathbf{A})$ is independent and identically distributed (IID) from $\mathcal{D}$, where $\mathbf{X}$ is the features of all nodes and $\mathbf{A}$ is the adjacency matrix. Following Miao et al. (2022), we assume that there exists a subgraph $\mathcal{G}^*$ such that the label $\mathbf{Y}$ for graph $\mathcal{G}$ is determined by $\mathbf{Y} = f(\mathcal{G}^*) + \epsilon$ for some $\epsilon$ as noise independent of $\mathcal{G}$, where $f$ is an invertible function projecting the set of subgraphs to the label space. Guided by the target of searching $\mathcal{G}^*$, current explainers mainly leverage feature attribution and selection to extract the subgraph (Wang et al., 2021).

**Feature attribution.** Current explainers mainly perform feature attribution by leveraging:

- Gradient-like signals *w.r.t.* the graph structure (Baldassarre & Azizpour, 2019; Pope et al., 2019). For example, SA (Baldassarre & Azizpour, 2019) directly calculates the gradients of GNN's loss *w.r.t.* adjacency matrix as the importance scores of edges;
- Attention scores of structural features (Luo et al., 2020; Miao et al., 2022). For example, GSAT (in its post-hoc working mode) (Miao et al., 2022) trains a parameterized predictor to generate the stochastic attention for each edge as their importance;
- Mask scores of structural features (Ying et al., 2019; Wang et al., 2021). For example, GNNExplainer (Ying et al., 2019) adds soft masks to the input features and trains them by maximizing the mutual information between the masked outcome and target prediction;
- Prediction changes on structure perturbations (Yuan et al., 2021; Lin et al., 2021). For example, PGMExplainer (Vu & Thai, 2020) collects the prediction change on the random node perturbations and learns a Bayesian network from these observations.

**Feature selection.** With attribution scores of features, input features are sequentially filled into the set of salient features to generate the explanatory subgraph according to their importance rank. Many regularization terms are introduced to guide this process. For example, sparsity constraints (Ying et al., 2019; Schlichtkrull et al., 2021) typically leverage the $l_1$ norm to guarantee that the selected subgraph remains within a prescribed size; connective constraints (Luo et al., 2020; Wang et al., 2021) give more selective probabilities to the edges connecting with the part selected already; more recently, information bottleneck constraints (Miao et al., 2022) are proposed to squeeze the mutual information between the input graph and the selected subgraph.

## 3 RETHINKING THE ROLE OF REGULARIZATION

In this section, we rethink the role of regularization in GNNs explainers. We start with a new form of graph information bottleneck tailored for explainability and the formulation of the feature attribution and selection (Section 3.1). Guided by the above theory, we analyze the effect of regularization in two phases, respectively (Section 3.2).

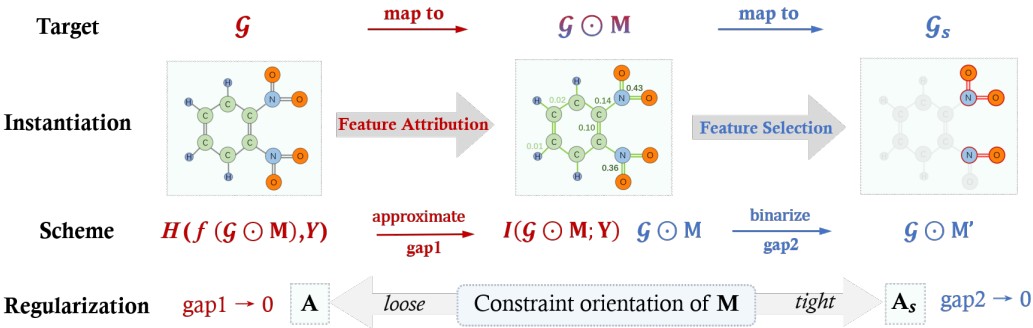

Figure 1: The role of regularization in the phases of feature attribution and feature selection.

## 3.1 GRAPH INFORMATION BOTTLENECK FOR EXPLAINABILITY

The principle of GIB is widely leveraged to guide the subgraph generation by prevailing GNNs explainers (Wu et al., 2020; Miao et al., 2022). The general formulation of GIB is shown as follows:

- **Definition 1 (GIB (Yu et al., 2021))** *Given an input graph $\mathcal{G}$ and its label $\mathbf{Y}$, GIB seeks for a maximally informative yet sparse subgraph by optimizing the following objective:*

$$\arg\max_{\mathcal{G}_s} I\left(\mathcal{G}_s; \mathbf{Y}\right) - \beta I\left(\mathcal{G}_s; \mathcal{G}\right), \text{ s.t. } \mathcal{G}_s \in \mathbb{G}_{sub}(\mathcal{G}), \quad (1)$$

*where $\mathbb{G}_{sub}(\mathcal{G})$ indicates the set of all subgraphs of $\mathcal{G}$ and $\beta$ is the Lagrangian multiplier.*

To instantiate the objective of GIB, considering the discreteness and non-differentiability of subgraph $\mathcal{G}_s$, existing explanation methods typically take $\mathcal{G} \odot \mathbf{M}$ as the proxy of $\mathcal{G}_s$, where $\mathbf{M}$ is the explanatory mask sharing the same size with $\mathbf{A}$. Concretely, for mask-based methods, $\mathbf{M}$ is the trainable masks; for attention-based methods, $\mathbf{M}$ is the dualization of the attention matrix; and for perturbation-based methods, $\mathbf{M}$ is the matrix recording corresponding features are perturbed or not.

In sight of this, we first replace $\mathcal{G}_s$ in Equation 1 with $\mathcal{G} \odot \mathbf{M}$. Moreover, according to the invariance of the mutual information (MI) to invertible transformation, we rewrite Equation 1 to introduce the new GIB form tailored for GNNs explainability. Detailed derivation is provided in Appendix A.1.

- **Definition 2 (GIBE)** *Given an input graph $\mathcal{G}$ and its label $\mathbf{Y}$, GIBE seeks for the explanatory mask $\mathbf{M}$ to generate the explanation $\mathcal{G} \odot \mathbf{M}$ by optimizing the following objective:*

$$\arg\max_{\mathbf{M}} \underbrace{I\left(\mathcal{G} \odot \mathbf{M}; \mathbf{Y}\right)}_{feature\ attribution} + \alpha \underbrace{\left[I\left(\mathcal{G} \odot \mathbf{M}; \mathcal{G}^*\right) - I\left(\mathcal{G} \odot \mathbf{M}; \mathcal{G}\right)\right]}_{feature\ selection}. \quad (2)$$

*where $\alpha$ is the tradeoff parameter which equals to $\beta/(1-\beta)$ for $\beta$ in Definition 1.*

Theoretically, employing Data Processing Inequality (DPI) (Cover & Thomas, 2006) along the Markov chain, $\mathcal{G}^* \to \mathcal{G} \to \mathbf{Y}$, the optimal solution $\mathbf{M}$ of Equation 2 can be proved to be equal to the adjacency matrix of $\mathcal{G}^*$. We provide detailed derivation in Appendix A.2.

Note that GIBE is the first attempt to in-depth combine the principle of GIB and the common paradigm of the post-hoc explainability (*i.e.*, feature attribution and selection). Specifically, the first term of Equation 2 is the optimization objective of feature attribution, which maps the information of feature importance in $\mathcal{G}$ to mask, $\mathcal{G} \mapsto \mathcal{G} \odot \mathbf{M}$; and the second term is the objective of feature selection, which maps the information in the mask to subgraph, $\mathcal{G} \odot \mathbf{M} \mapsto \mathcal{G}_s$. In conclusion, GIBE endows the GIB with better ability to directly guide the construction of the explanation methods.

## 3.2 REGULARIZATION IN TWO PHASES

**Regularization in feature attribution.** Directly calculating the training objective of feature attribution, $I\left(\mathcal{G} \odot \mathbf{M}; \mathbf{Y}\right)$ is tough since the latent distribution $\mathbb{P}(\mathcal{G} \odot \mathbf{M}, \mathbf{Y})$ is notoriously intractable. Hence, existing explainers typically introduce the parameterized variational approximation $\mathbb{P}_\theta\left(\mathbf{Y} \mid \mathcal{G} \odot \mathbf{M}\right)$ for $\mathbb{P}\left(\mathbf{Y} \mid \mathcal{G} \odot \mathbf{M}\right)$ according to:

$$I\left(\mathcal{G} \odot \mathbf{M}; \mathbf{Y}\right) = \mathbb{E}_{\mathcal{G} \odot \mathbf{M}, \mathbf{Y}}\left[\log \mathbb{P}_\theta\left(\mathbf{Y} \mid \mathcal{G} \odot \mathbf{M}\right)\right] + H(\mathbf{Y})$$
$$+ \mathbb{E}_{\mathcal{G} \odot \mathbf{M}}\left[\mathrm{KL}\left(\mathbb{P}\left(\mathbf{Y} \mid \mathcal{G} \odot \mathbf{M}\right) \| \mathbb{P}_\theta\left(\mathbf{Y} \mid \mathcal{G} \odot \mathbf{M}\right)\right)\right]. \quad (3)$$

Detailed derivation is provided in Appendix A.3. Treating classifier $f'$ as the proxy function of $\mathbb{P}_\theta$, the first term of Equation 3 is specified as the cross-entropy between $\mathbf{Y}$ and $f'(\mathcal{G} \odot \mathbf{M})$. In this case, since the second term $H(\mathbf{Y})$ is a constant, the gap between $I(\mathcal{G} \odot \mathbf{M}; \mathbf{Y})$ and above cross-entropy, shown as *gap1* in Figure 1, solely depends on the third term of Equation 3. Note that this term is also the Kullback-Leibler (KL) Divergence between $\mathbb{P}(\mathbf{Y} \mid \mathcal{G} \odot \mathbf{M})$ and $f'(\mathcal{G} \odot \mathbf{M})$.

To bridge this gap and maximize the effectiveness of feature attribution, regularization should play a role in constraining $\mathbb{P}(\mathbf{Y} \mid \mathcal{G} \odot \mathbf{M})$ to $f'$. Since $f'$ is trained for data distribution $\mathbb{P}(\mathcal{G}, \mathbf{Y})$, this role transfers to constrain $\mathcal{G} \odot \mathbf{M}$ to $\mathcal{G}$. In this case, the constraint orientation of $\mathbf{M}$ should be loosened to $\mathbf{A}$ for squeezing gap in feature attribution to 0, as shown in the left side of Figure 1.

**Regularization in feature selection.** Feature selection typically binarizes $\mathbf{M}$ to achieve the mapping $\mathcal{G} \odot \mathbf{M} \mapsto \mathcal{G}_s$. To bridge the gap between $\mathbf{M}$ and its binaryzation $\mathbf{M}'$, shown as *gap2* in Figure 1, the optimization objective of feature selection, $I(\mathcal{G} \odot \mathbf{M}; \mathcal{G}^*) - I(\mathcal{G} \odot \mathbf{M}; \mathcal{G})$ in Equation 2 is usually maximized by: (1) discrete constraint (Ying et al., 2019) and connectivity constraint (Luo et al., 2020), which are work for maximizing $I(\mathcal{G} \odot \mathbf{M}; \mathcal{G}^*)$; (2) sparsity constraint (Schlichtkrull et al., 2021), which leverages the $l_1$ or $l_2$ norm to minimize $I(\mathcal{G} \odot \mathbf{M}; \mathcal{G})$.

That is, in the phase of feature selection, regularization plays a role in constraining $\mathcal{G} \odot \mathbf{M}$ to $\mathcal{G}_s$. In other words, the constraint orientation of $\mathbf{M}$ should be tightened to $\mathbf{A}_s$ (*i.e.*, adjacency matrix of $\mathcal{G}_s$) for squeezing the gap in feature selection to 0, as shown in the right side of Figure 1.

In conclusion, while the constraint orientation of $\mathbf{M}$ is beneficial to feature attribution and the gap in feature attribution tends to zero, the gap in feature selection will inevitably become large, and vice versa. In other words, since the role of regularization is totally different across two phases, regularization is essentially a tradeoff to guarantee the mapping effectiveness both in two phases:

- **Proposition 1 (Essence of Regularization)** *Regularization in GNNs explainability is essentially the tradeoff scheme to pursue the balance between the phases of feature attribution and selection.*

## 4 PROPOSITIONS OF REGULARIZATION IN EXPLANATION METHODS

In this section, we derive three intriguing propositions stemming from GIBE and *Proposition 1*, which respectively reveal the relation between regularization and three important concepts in GNNs explainability (*i.e.*, sparsity (Ying et al., 2019; Lucic et al., 2022), stochastic mechanism (Luo et al., 2020; Wang et al., 2021) and OOD issue (Miao et al., 2022; Wu et al., 2022)). Inspired by these propositions, we propose two simple yet effective regularization schemes.

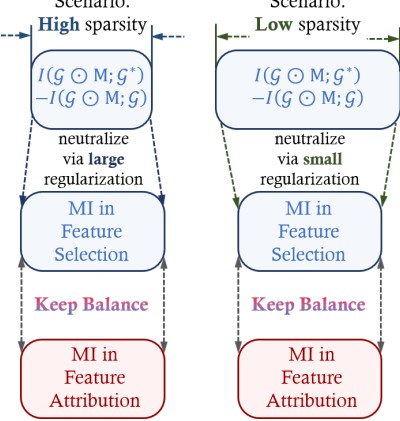

Figure 2: The rationale of sparsity adaptive regularization scheme.

### 4.1 REGULARIZATION & SPARSITY

We first focus on the relationship between regularization and predefined sparsity $\eta$ of targeted subgraph $\mathcal{G}^*$, where $\eta = 1 - |\mathcal{G}^*|/|\mathcal{G}|$. According to Equation 2, sparsity variation of $\mathcal{G}^*$ only affect the objective of feature selection, $I(\mathcal{G} \odot \mathbf{M}; \mathcal{G}^*) - I(\mathcal{G} \odot \mathbf{M}; \mathcal{G})$. Therefore, we aim to derive the relation between $\eta$ and the phase of feature selection in current explanation methods. The process of derivation is exhibited as follows:

To calculate this objective, we first introduce the variational approximation $\mathbb{P}_\phi(\mathcal{G} \odot \mathbf{M} \mid \mathcal{G}^*)$ for $\mathbb{P}(\mathcal{G} \odot \mathbf{M} \mid \mathcal{G}^*)$ and $\mathbb{Q}$ for marginal distribution $\mathbb{P}(\mathcal{G} \odot \mathbf{M})$ simultaneously. Then the lower bound of the objective can be derived as:

$$
\begin{aligned}
I(\mathcal{G} \odot \mathbf{M}; \mathcal{G}^*) - I(\mathcal{G} \odot \mathbf{M}; \mathcal{G}) \geq{} & \mathbb{E}_{\mathcal{G} \odot \mathbf{M}, \mathcal{G}^*}\left[\log \mathbb{P}_\phi(\mathcal{G} \odot \mathbf{M} \mid \mathcal{G}^*)\right] + H(\mathcal{G} \odot \mathbf{M}) \\
& - \mathbb{E}_{\mathcal{G} \odot \mathbf{M}, \mathcal{G}}\left[\mathrm{KL}\left(\mathbb{P}(\mathcal{G} \odot \mathbf{M} \mid \mathcal{G}) \| \mathbb{Q}(\mathcal{G} \odot \mathbf{M})\right)\right].
\end{aligned}
\tag{4}
$$

Detailed derivation is shown in Appendix A.4. While $\mathbf{M}$ is inherited from feature attribution, the above lower bound is in positive relation to its first term, $\mathbb{E}_{\mathcal{G} \odot \mathbf{M}, \mathcal{G}^*}\left[\log \mathbb{P}_\phi(\mathcal{G} \odot \mathbf{M} \mid \mathcal{G}^*)\right]$ since the rest of terms are constants. To calculate its first term, the variational distribution $\mathbb{P}_\phi(\mathcal{G} \odot \mathbf{M} \mid \mathcal{G}^*)$ is defined as follow: for every two directed node pair $(u, v)$ in $\mathcal{G}^*$, we sample the elements of $\mathbf{M}$

by $m_{uv} \sim \text{Bern}(z)$ where $z \in [0,1]$ is a hyperparameter. Borrowing the above $\mathbb{P}_\phi$, the first term of Equation 4 is equal to:

$$\mathbb{E}_{\mathcal{G} \odot \mathbf{M}, \mathcal{G}^*} \left[ \log \mathbb{P}_\phi \left( \mathcal{G} \odot \mathbf{M} \mid \mathcal{G}^* \right) \right] = \sum_{(u,v) \in \mathcal{E}^*} m_{uv} \log \frac{m_{uv}}{z} + (1 - m_{uv}) \log \frac{(1 - m_{uv})}{1 - z}, \quad (5)$$

where $\mathcal{E}^*$ is the set of edges in $\mathcal{G}^*$. Note that the sum in Equation 5 has positive relation to the number of terms $|\mathcal{E}^*|$, which equals to $(1 - \eta)|\mathcal{G}|$, since $\mathbf{M}$ is inherent and the values of every terms are fixed. That is, in the phase of feature selection, the lower bound of the objective is in negative relation to $\eta$. It is worth to mention that Equation 5 is similar with the regularization terms proposed in GSAT (Miao et al., 2022). However, it is appropriate for mainly post-hoc explanation methods while terms proposed in GSAT is tailored for itself solely.

Therefore, for an interpreting task appealing to low sparsity, the magnitude of objective in feature selection is large. Since regularization is the tradeoff scheme between two phases (*Proposition 1*), it should be loosened to a favorable direction for feature attribution to keep the balance, and vise versa. We illustrate this process in Figure 2. More formally:

- **Proposition 2 (Sparsity-adaptive Regularization)** *For a certain interpreting task, let $\mathbf{K}_i$ and $\mathbf{K}_j$ are the optimal coefficient vector of regularization under the predefined explanatory sparsity $\eta_i$ and $\eta_j$. For all $\eta \in (0,1)$ we have:*

$$\eta_i \geq \eta_j \Leftrightarrow \mathbf{K}_i \geq \mathbf{K}_j \quad (6)$$

According to *Proposition 2*, we propose a simple yet effective scheme called Sparsity-adaptive Regularization Scheme (SRS) to enhance the performance of existing explainers. Concretely, SRS first seeks for the optimal coefficients $\mathbf{K}_i$ under the certain sparsity $\eta_i$ by grid search, then for other sparsity $\eta_j$, SRS sets $\mathbf{K}_j$ according to $\mathbf{K}_j = \frac{\eta_j}{\eta_i} \mathbf{K}_i$. Specific implementation and experimental verification are provided in Section 5.2.

## 4.2 REGULARIZATION & STOCHASTIC MECHANISM

In this part we focus on the relationship between the regularization and the stochastic mechanism. Recent years have witnessed a surge in research that focuses on leveraging the stochastic mechanism to enhance the performance of GNNs explainability (Luo et al., 2020; Wang et al., 2021; Miao et al., 2022). Concretely, the probability of graph $\mathcal{G}$ is first factorized as $P(\mathcal{G}) = \Pi_{(i,j) \in \mathcal{E}} P(e_{ij})$, where $e_{ij} = 1$ if the edge $(i,j)$ exists, and 0 otherwise. Then the Bernoulli distribution is employed to instantiate $P(e_{ij})$, where the stochastic mechanism is introduced in this process (Luo et al., 2020). Meanwhile, the gumbel-softmax reparameterization trick is applied to make sure the gradient in the Bernoulli distribution is traceable (Jang et al., 2017). However, despite the satisfactory performance of the stochastic mechanism, current researches mainly neglected the theoretical basis of the stochastic mechanism.

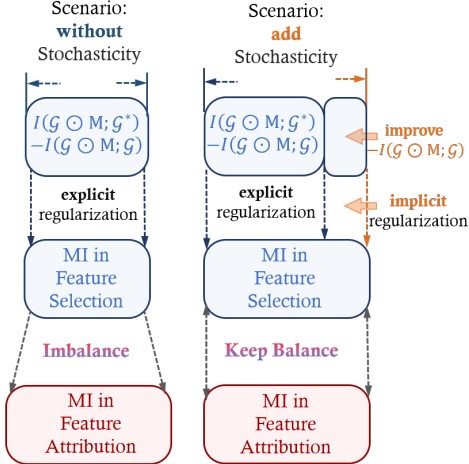

Figure 3: Rationale of stochastic mechanism in explainability: an implicit regularization term.

We argue that the stochastic mechanism bolsters the explanation methods via facilitating the feature selection. Specifically, employing the stochastic mechanism to the training stage of $\mathbf{M}$ can be regarded as adding random noise to $\mathbf{M}$ (Shwartz-Ziv & Tishby, 2017). The entropy of the mask's distribution will continue to increase during this process under the constraints of the information bottleneck. That in turn maximizes the conditional entropy $H(\mathcal{G}|\mathcal{G} \odot \mathbf{M})$. Since the entropy of input graph $H(\mathcal{G})$ is a constant, mutual information $I(\mathcal{G} \odot \mathbf{M}, \mathcal{G}) = H(\mathcal{G}) - H(\mathcal{G}|\mathcal{G} \odot \mathbf{M})$ is minimized contributed to stochastic mechanism. In conclusion, the stochastic mechanism can accelerate the process of minimizing $I(\mathcal{G} \odot \mathbf{M}, \mathcal{G})$, thus endowing the compression of the explanatory subgraph. That is, it can be regarded as the implicit regularization term in favor of the phase of feature selection. We summarize this rationale as *Proposition 3* and verify it in Section 5.3.

- **Proposition 3 (Rationale of Stochastic Mechanism)** *Stochastic mechanism works as the implicit regularization term, which endows GNNs explainability with better compressibility.*

## 4.3 REGULARIZATION & OOD

Many recent endeavors have been made towards revealing the out-of-distribution issue of the post-hoc explanation methods (Chang et al., 2019; Qiu et al., 2022; Wu et al., 2022; Miao et al., 2022). Concretely, OOD issue is posed in the data space since the distribution of full graph $\mathbb{P}(\mathcal{G})$ differs from that of subgraph $\mathbb{P}(\mathcal{G}_s)$ *w.r.t.* some properties of graph data such as size (Bevilacqua et al., 2021), degree (Tang et al., 2020) and homophily (Lei et al., 2022). Thus it is fallacious to leverage the original GNN $f'$, which is trained for the full graphs, to estimate the MI in terms of subgraphs. Unfortunately, most previous explainers (Ying et al., 2019; Luo et al., 2020; Vu & Thai, 2020; Miao et al., 2022) simply feed $\mathcal{G} \odot \mathbf{M}$ into $f'$, unaware of that $I\left(f(\mathcal{G} \odot \mathbf{M}); \mathbf{Y}\right)$ is not proportional to $I\left(\mathcal{G} \odot \mathbf{M}; \mathbf{Y}\right)$, sometimes not even close.

To reveal the relationship between OOD issue and regularization, we scrutinize the OOD between $\mathbb{P}(\mathcal{G})$ and $\mathbb{P}(\mathcal{G}_s)$ and decouple it into two phases: (1) OOD-AT between $\mathbb{P}(\mathcal{G})$ and $\mathbb{P}(\mathcal{G} \odot \mathbf{M})$ posed in feature attribution and (2) OOD-SE between $\mathbb{P}(\mathcal{G} \odot \mathbf{M})$ and $\mathbb{P}(\mathcal{G}_s)$ posed in feature selection.

**OOD-AT.** We first focus on OOD-AT posed in feature attribution. According to Equation 3, feature attribution searches for the optimal $\mathbf{M}$ via calculating the cross-entropy $H(f(\mathcal{G} \odot \mathbf{M}), \mathbf{Y})$ to approximate $I\left(\mathcal{G} \odot \mathbf{M}; \mathbf{Y}\right)$. The approximation error, also be illustrated as *gap1* in Figure 1, is given as the sum of second and third term of Equation 3. For the second term, $H(\mathbf{Y})$, since it is the constant across different candidates $\mathbf{M}$, it will not affect the search results. Unfortunately, for the third term, $\mathrm{KL}\left(\mathbb{P}\left(\mathbf{Y} \mid \mathcal{G} \odot \mathbf{M}\right) \| \mathbb{P}_\theta\left(\mathbf{Y} \mid \mathcal{G} \odot \mathbf{M}\right)\right)$, OOD-AT makes this part far away from zero and inevitably fluctuate across different $\mathbf{M}$.

This fluctuation degrades the fairness of the searching process of optimal $\mathbf{M}$, and further degenerates the effectiveness of the mapping, $\mathcal{G} \mapsto \mathcal{G} \odot \mathbf{M}$ in feature attribution. Thus we formulate the tangible impact caused by OOD-AT as:

$$\text{OOD-AT} \propto \mathbb{D}_{\mathbf{M}}[\mathrm{KL}\left(\mathbb{P}\left(\mathbf{Y} \mid \mathcal{G} \odot \mathbf{M}\right) \| \mathbb{P}_\theta\left(\mathbf{Y} \mid \mathcal{G} \odot \mathbf{M}\right)\right)], \tag{7}$$

where $\mathbb{D}$ is the Variance. If regularization pushes $\mathbb{P}(\mathcal{G} \odot \mathbf{M})$ close to $\mathbb{P}(\mathcal{G})$ via constraining $\mathbf{M}$ to $\mathbf{A}$, the Equation 7 will close to zero and OOD-AT will cause little degradation. Note that this process is similar to the left side of Figure 1: the constrain orientation is loosened to $\mathbf{A}$ to eliminate the gap.

**OOD-SE.** We then focus on OOD-SE posed in feature selection. While feature selection binarizes $\mathbf{M}$ to get $\mathbf{M}'$ and generates $\mathcal{G}_s = \mathcal{G} \odot \mathbf{M}'$, the degree of OOD-SE can be simplified as the distance between $\mathbf{M}$ and $\mathbf{A}_s$. That is, OOD-SE issue will be remedied if regularization pushes $\mathbb{P}(\mathcal{G} \odot \mathbf{M})$ close to $\mathbb{P}(\mathcal{G}_s)$ via constraining $\mathbf{M}$ to $\mathbf{A}_s$.

In conclusion, just similar to the dilemma of regularization between two phases described in Section 3.2, the optimal regularization scheme for remedying OOD-AT and OOD-SE is totally different. That is, for a certain interpreting task, there is inherent uncooperativeness between OOD in feature attribution and OOD in feature selection. Note that the above analysis also explains why the OOD issue is the inherent limitation of post-hoc explainability: since we can only remedy one side and ignore the other side, OOD can not be thoroughly settled by adjusting the training objective and hyperparameters solely. We formulate this proposition as:

- **Proposition 3 (Rationale of OOD)** *For a certain interpreting task, the OOD issue is imputed to the contradictory effects of regularization on the phases of feature attribution and selection.*

Even though the OOD issue is inherent, we introduce a simple yet effective regularization scheme called OOD-resistant Regularization Scheme (ORS) to alleviate the above dilemma to some extent. Specifically, in the early stage of training, loose regularization of $\mathbf{M}$ is performed in favor of alleviating OOD-AT for the more precise mapping $\mathcal{G} \mapsto \mathcal{G} \odot \mathbf{M}$. Similarly, we then gradually tighten the regularization of $\mathbf{M}$ to alleviate the OOD-SE for the mapping $\mathcal{G} \odot \mathbf{M} \mapsto \mathcal{G}_s$. ORS can achieve better explanation performance than the common invariable regularization scheme. Specific implementation and experimental verification are provided in Section 5.3.

## 5 EXPERIMENT

In this section, we conduct extensive experiments to answer the following questions:

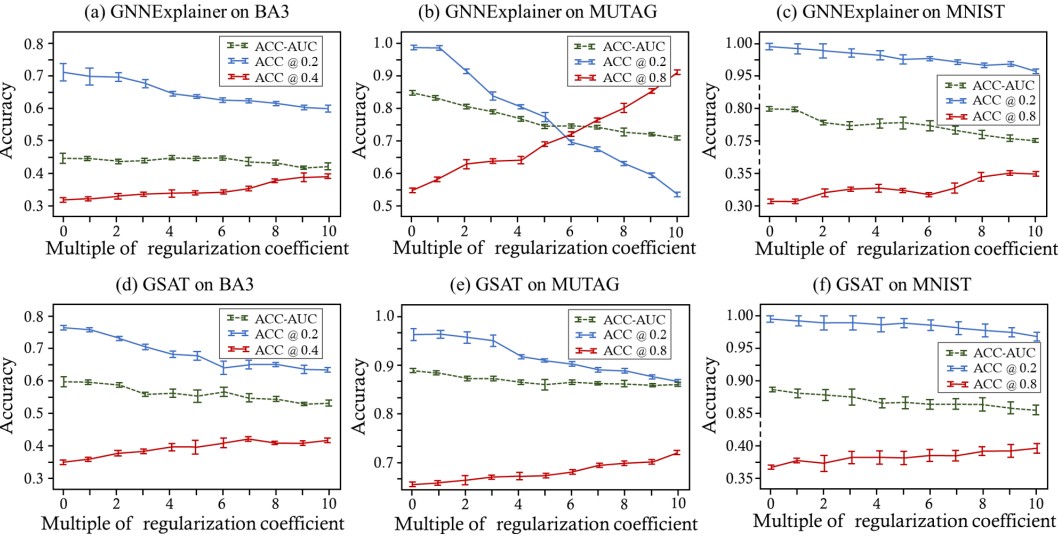

Figure 4: The performance of baseline explainers averaged across 10 runs for different sparsity, while the coefficient of regularization is changed. Best viewed in color.

- **RQ1:** Can *Proposition 1* provided in Section 3.2 and *Proposition 2* provided in Section 4.1 be verified? Furthermore, how does the SRS perform?
- **RQ2:** Can *Proposition 3* provided in Section 4.2 and *Proposition 4* provided in Section 4.3 be verified? Furthermore, how does the ORS perform?

## 5.1 SETTINGS

**Datasets and Target GNNs.** We use one synthetic dataset and two real datasets which are publicly accessible and train three popular GNN models for them. More details can be found in Appendix B.

- **BA-3motifs** (Ying et al., 2019) contains 3,000 graphs attaching with the motif types: house, cycle, or grid, for which a trained ASAP (Ranjan et al., 2020), has achieved a 99.3% testing accuracy.
- **MUTAG** (Kazius et al., 2005) contains 4,337 molecule graphs categorized into two classes based on their mutagenic effect on the Gram-negative bacterium. The well-trained Graph Isomorphism Network (GIN) (Xu et al., 2019) has achieved a 97.7% testing accuracy.
- **MNIST** (Monti et al., 2017) superpixel converts 70,000 images into the graphs of superpixel adjacency. A trained Spline-based GNN (Fey et al., 2018) has achieved a 94.6% testing accuracy.

**Evaluation Metrics and Baselines.** We select three commonly used metrics to evaluate our results: Accuracy, Precision, and Fidelity. Moreover, we leverage six state-of-the-art methods to verify proposed propositions and schemes, covering the GNNExplainer (Ying et al., 2019), PGExplainer (Luo et al., 2020), GraphMask (Schlichtkrull et al., 2021), CF-GNNExplainer (Lucic et al., 2022), Refine (Wang et al., 2021) and GSAT (Miao et al., 2022). More details is provided in Appendix B.

## 5.2 EVALUATION OF PROPOSITION 1, PROPOSITION 2 AND SRS (RQ1)

We first focus on verifying the *Proposition 1* and *Proposition 2* provided in Section 3.2 and Section 4.1 simultaneously. Since the role of regularization is to keep the balance between two phases (*Proposition 1*), the variation of its coefficient $\mathbf{K}$ will break the balance and ulteriorly impose opposite impacts on the performance under different sparsity $\eta$ (*Proposition 2*).

In sight of this, we start with the optimal coefficients of regularization for ACC-AUC by grid search, then we deliberately increase $\mathbf{K}$ and record the accuracy under different $\eta$. Experiment results are shown in Figure 4, where the horizontal axis represents the multiple of the increment of $\mathbf{K}$ and the length of marker shows the variance. Note that the performance of other baselines have similar trends to the results in Figure 4.

Table 1: The performance of baseline explainers averaged across 10 runs. The best performing methods are bold with blue lines, and the strongest baselines are underlined.

| | BA3-motif | | | Mutagenicity | | MNIST | | |
|---|---|---|---|---|---|---|---|---|
| | ACC-AUC | Fidelity@0.5 | Precision@5 | ACC-AUC | Fidelity@0.5 | ACC-AUC | Fidelity@0.5 | Precision@10 |
| GNNExplainer | $0.439_{\pm0.024}$ | $0.389_{\pm0.034}$ | $0.532_{\pm0.031}$ | $0.841_{\pm0.022}$ | $0.442_{\pm0.036}$ | $0.798_{\pm0.047}$ | $0.274_{\pm0.027}$ | $0.512_{\pm0.029}$ |
| +SRS | $0.497_{\pm0.016}$ | $0.413_{\pm0.018}$ | $0.597_{\pm0.020}$ | $0.897_{\pm0.024}$ | $0.503_{\pm0.032}$ | $0.805_{\pm0.044}$ | $0.295_{\pm0.008}$ | $0.542_{\pm0.021}$ |
| PGExplainer | $0.511_{\pm0.032}$ | $0.482_{\pm0.022}$ | $0.564_{\pm0.040}$ | $0.809_{\pm0.055}$ | $0.547_{\pm0.021}$ | $0.800_{\pm0.050}$ | $0.280_{\pm0.011}$ | $0.528_{\pm0.027}$ |
| +SRS | $0.558_{\pm0.024}$ | $\mathbf{0.502}_{\pm0.017}$ | $0.617_{\pm0.025}$ | $0.843_{\pm0.038}$ | $0.586_{\pm0.022}$ | $0.822_{\pm0.041}$ | $0.304_{\pm0.006}$ | $0.570_{\pm0.020}$ |
| GraphMask | $0.489_{\pm0.051}$ | $0.408_{\pm0.027}$ | $0.579_{\pm0.037}$ | $0.826_{\pm0.056}$ | $0.400_{\pm0.030}$ | $0.817_{\pm0.047}$ | $0.291_{\pm0.019}$ | $0.437_{\pm0.032}$ |
| +SRS | $0.524_{\pm0.040}$ | $0.444_{\pm0.022}$ | $0.594_{\pm0.014}$ | $0.878_{\pm0.013}$ | $0.451_{\pm0.025}$ | $0.839_{\pm0.039}$ | $0.315_{\pm0.012}$ | $0.499_{\pm0.020}$ |
| CF-GNNExplainer | $0.453_{\pm0.044}$ | $0.416_{\pm0.030}$ | $0.508_{\pm0.032}$ | $0.842_{\pm0.038}$ | $0.474_{\pm0.027}$ | $0.822_{\pm0.037}$ | $0.221_{\pm0.013}$ | $0.507_{\pm0.021}$ |
| +SRS | $0.476_{\pm0.032}$ | $0.445_{\pm0.014}$ | $0.549_{\pm0.027}$ | $0.874_{\pm0.039}$ | $0.496_{\pm0.030}$ | $0.852_{\pm0.028}$ | $0.260_{\pm0.008}$ | $0.556_{\pm0.020}$ |
| Refine | $0.542_{\pm0.031}$ | $0.447_{\pm0.025}$ | $0.598_{\pm0.019}$ | $\underline{0.905}_{\pm0.055}$ | $0.579_{\pm0.034}$ | $0.854_{\pm0.062}$ | $0.300_{\pm0.016}$ | $\underline{0.536}_{\pm0.022}$ |
| +SRS | $0.550_{\pm0.023}$ | $0.472_{\pm0.011}$ | $\mathbf{0.632}_{\pm0.026}$ | $0.921_{\pm0.041}$ | $0.593_{\pm0.016}$ | $0.870_{\pm0.034}$ | $0.333_{\pm0.014}$ | $\mathbf{0.574}_{\pm0.018}$ |
| GSAT | $\underline{0.582}_{\pm0.027}$ | $0.477_{\pm0.035}$ | $\underline{0.602}_{\pm0.022}$ | $0.892_{\pm0.031}$ | $\underline{0.581}_{\pm0.022}$ | $\underline{0.871}_{\pm0.042}$ | $\underline{0.322}_{\pm0.012}$ | $0.520_{\pm0.015}$ |
| +SRS | $\mathbf{0.604}_{\pm0.016}$ | $0.490_{\pm0.023}$ | $0.629_{\pm0.019}$ | $0.913_{\pm0.023}$ | $\mathbf{0.609}_{\pm0.029}$ | $\mathbf{0.890}_{\pm0.033}$ | $\mathbf{0.342}_{\pm0.010}$ | $0.542_{\pm0.020}$ |
| **Relative Impro.** | **4.2**% | **7.2**% | **5.3**% | **6.1**% | **7.9**% | **3.4**% | **5.8**% | **8.0**% |

According to Figure 4 we have the following observations:

- While **K** is changed, the fluctuation of overall performance (*i.e.*, ACC-AUC) is not too strong ($\pm7.25\%$) on average. However, the fluctuation of ACC in high sparsity and low sparsity is obviously larger ($\pm21.86\%$) than that of ACC-AUC. These observations verify the role of regularization: a tradeoff to keep the balance and guarantee the overall performance at a high level.
- ACC under low sparsity and ACC-AUC are decreasing with the increment of the regularization coefficient. On the contrary, ACC in high sparsity is increasing by a large margin ($29.63\% \uparrow$) on average. These increases up to a maximum of $67.83\%$ in MUTAG are counterintuitive and eccentric. Whereas, these observations exactly conform to the theoretical analysis in *Propositon 2*: loose constraint (*i.e.*, large **K**) is beneficial to the performance in high sparsity, while tight constraint (*i.e.*, small **K**) is in favor of the performance under low sparsity.

While the above observations verify the *Proposition 1* and *Proposition 2*, we now evaluate the effectiveness of the derived scheme, SRS. Specifically, for a certain interpreting task, we first seek for the optimal **K** under $\eta = 0.5$ by grid search, then we reduce or enlarge **K** proportionally to the variation of $\eta$ in $(0.1, ..., 0.9)$. Results are summarized in Table 1. We have the following observations:

- The baseline explainers enhanced by SRS outperform themselves in all cases. To be more specific, SRS achieves significant improvements over the six baselines *w.r.t.* fidelity by 7.9% and 7.2% averagely in MUTAG and BA3-motif, respectively. The improvement is up to a maximum of 12.8% when GNNExplainer is employed to interpret graphs in MUTAG. It demonstrates the effectiveness and universality of SRS, and verifies that SRS can be leveraged to boost the accuracy of current explainers. We attribute these improvements to the ability that SRS can allocate the most adaptive coefficient of regularization for diverse explanatory sparsity.
- SRS provides much stabler explanations than the baselines for the much smaller variance. More specifically, STD of SRS outperforms baseline by a larger margin ($22.8\% \downarrow$) on average. This phenomenon verifies that the performance of most explainers is not stable since it depends on the matching degree between the task and regularization terms, while SRS can avoid this dilemma by adaptively allocating regularization terms.

## 5.3 EVALUATION OF PROPOSITION 3, PROPOSITION 4 AND ORS (RQ2)

We now focus on verifying the *Proposition 3* proposed in Section 4.2 and the *Proposition 4* proposed in Section 4.3. According to *Proposition 3*, the stochastic mechanism endows GNNs explainability with better compressibility, which has a similar rationale to the regularization term. In sight of this, we gradually increase the magnitude of stochasticity in explainers and observe whether the performance has the same trends as the results in Figure 4. The overall results are summarized in Figure 5. We have the following observations:

- The ACC under low sparsity and ACC-AUC are decreasing ($8.76\% \downarrow$ and $19.85\% \downarrow$) with the increment of stochasticity magnitude, while the ACC under high sparsity is increasing ($14.52\% \uparrow$) in this case. These increases up to a maximum of $37.03\%$ in MUTAG. Therefore, the large magnitude of stochasticity is beneficial to the performance in high sparsity and vice versa. This effect is similar to the effect of regularization terms. In conclusion, the same trends between Figure 4

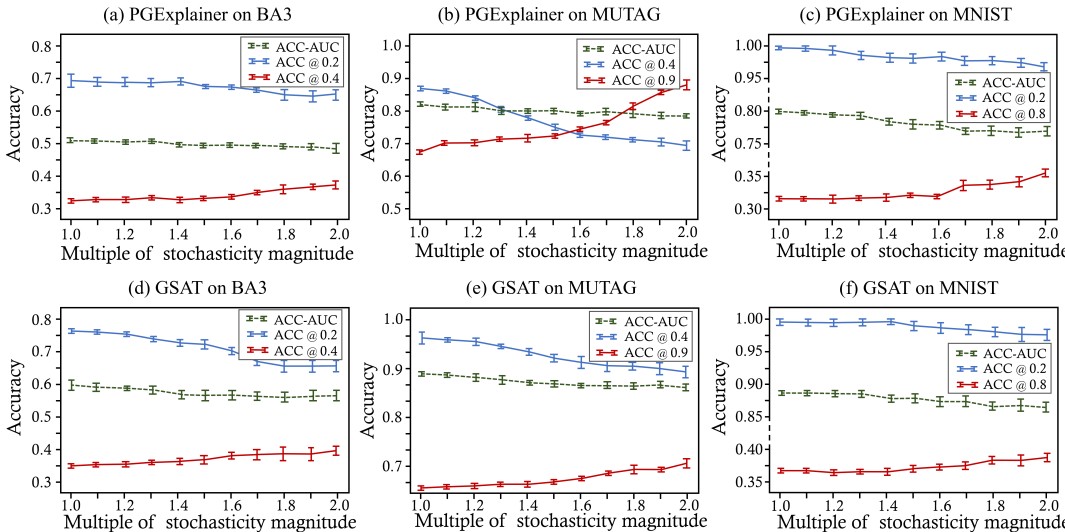

Figure 5: The performance of baseline explainers averaged across 10 runs for different sparsity while the magnitude of stochasticity is changed. Best viewed in color.

and Figure 5 verify that the stochastic mechanism has a similar rationale with the regularization terms, and further verify the *Proposition 3*.

In terms of *Proposition 4*, since the degree of OOD is intractable, we directly leverage the derived scheme: ORS to remedy the OOD issue in explainers. We surmise the results in Table 2. We find that:

- The explainers enhanced by ORS get the higher accuracy on the graph classification tasks than themselves. Specifically, ORS achieves significant improvements over the strongest baselines *w.r.t.* ACC-AUC by 4.1% and 3.5% in MUTAG and BA3-motif, respectively. These improvements verify the reliability and effectiveness of the ORS. We contribute these improvements mainly to the advantage of remedying the OOD issue in the phases of feature attribution and selection. This observation further validates the theoretical analysis of the OOD issue in the above two phases (*Proposition 4*) indirectly.

Table 2: The performance of baseline explainers averaged across 10 runs. The best performing methods are bold with blue line, and the strongest baselines are underlined.

| | BA3-motif | Mutagenicity | MNIST |
|---|---|---|---|
| GNNExplainer | $0.439_{\pm0.024}$ | $0.841_{\pm0.022}$ | $0.798_{\pm0.047}$ |
| +ORS | $0.466_{\pm0.033}$ | $0.860_{\pm0.031}$ | $0.814_{\pm0.054}$ |
| PGExplainer | $0.511_{\pm0.032}$ | $0.809_{\pm0.055}$ | $0.800_{\pm0.050}$ |
| +ORS | $0.537_{\pm0.019}$ | $0.821_{\pm0.026}$ | $0.813_{\pm0.027}$ |
| GraphMask | $0.489_{\pm0.051}$ | $0.826_{\pm0.056}$ | $0.817_{\pm0.047}$ |
| +ORS | $0.496_{\pm0.042}$ | $0.844_{\pm0.011}$ | $0.828_{\pm0.053}$ |
| CF-GNNExplainer | $0.453_{\pm0.044}$ | $0.842_{\pm0.038}$ | $0.822_{\pm0.037}$ |
| +ORS | $0.470_{\pm0.046}$ | $0.853_{\pm0.029}$ | $0.844_{\pm0.025}$ |
| Refine | $0.542_{\pm0.031}$ | $\underline{0.905}_{\pm0.055}$ | $0.854_{\pm0.062}$ |
| +ORS | $0.551_{\pm0.017}$ | $\mathbf{0.911}_{\pm0.018}$ | $0.866_{\pm0.030}$ |
| GSAT | $\underline{0.582}_{\pm0.027}$ | $0.892_{\pm0.031}$ | $\underline{0.871}_{\pm0.042}$ |
| +ORS | $\mathbf{0.591}_{\pm0.023}$ | $0.903_{\pm0.035}$ | $\mathbf{0.883}_{\pm0.017}$ |
| **Relative Impro.** | **3.3%** | **3.6%** | **3.1%** |

## 6 CONCLUSION

In this work, we rethink the role of regularization in GNNs explainability from the perspective of information theory. We first retrospect the concept of GIB and derive the new GIB form tailored for GNNs explainability: GIBE. The role of regularization in the phases of feature attribution and selection are explored respectively under the guidance of GIBE. Moreover, four intriguing propositions and two common optimization schemes of regularization are introduced inspired by the above insight. Extensive experiments are conducted on both synthetic and real-world datasets to validate the rationality of our propositions and the superiority of our scheme. This work represents an initial attempt of digging deeper into the regularization in explainability theoretically, which provides a new perspective to understand the role of regularization.

## ETHICS STATEMENT

In this work, we rethink the role of regularization in GNNs explainability from the perspective of information theory, where no human subject is related. We believe digging deeper into the regularization theoretically is beneficial for performing the better explanations and consequently improving the model's transparency in real-world applications.

## REPRODUCIBILITY

We summarize the efforts made to ensure reproducibility in this work. (1) Datasets: We use one synthetic dataset and two real datasets which are publicly accessible, where the processing details are included in Appendix B. (2) Model Training: We provide the training details (including hyper-parameter settings) in Appendix B which are consistent with our implementation in the code (*cf.* the anonymous link provided in Abstract). (3) Theoretical Results: All assumptions and proofs can be referred to Appendix A.

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

## A  THE PROCESS OF DERIVATION IN TERMS OF INFORMATION THEORY

### A.1  DERIVATION OF GIBE

We focus on the detailed derivation of GIBE. Firstly, the proportion of mutual information: *invariance to invertible transformations* (Shwartz-Ziv & Tishby, 2017) can be formulated as:

$$I(X;Y) = I(\psi(X);\phi(Y))),\tag{8}$$

for any invertible functions $\psi$ and $\phi$. Since $\mathbf{Y}$ is determined by $\mathcal{G}^*$ in the sense that $\mathbf{Y} = f(\mathcal{G}^*) + \epsilon$ for some deterministic invertible function $f$, according to Equation 8, $I(\mathcal{G}_s;\mathbf{Y})$ in the objective of GIB (*i.e.*, Equation 1) can be rewrote as (Miao et al., 2022):

$$I(\mathcal{G}_s;\mathbf{Y}) = I(\mathcal{G}_s;\mathcal{G}^*).\tag{9}$$

Thus we can decompose the first term in the objective of GIB into:

$$I(\mathcal{G}_s;\mathbf{Y}) = (1-\beta)I(\mathcal{G}_s;\mathbf{Y}) + \beta I(\mathcal{G}_s;\mathcal{G}^*),\tag{10}$$

for any $\beta \in (0,1)$. Comparing Equation 10 and Equation 1, the objective of GIB can be rewritten as:

$$\underset{\mathcal{G}_s}{\arg\max}(1-\beta)I(\mathcal{G}_s;\mathbf{Y}) + \beta\left[I(\mathcal{G}\odot\mathbf{M};\mathcal{G}^*) - I(\mathcal{G}\odot\mathbf{M};\mathcal{G})\right].\tag{11}$$

We then substitute the coefficient $\alpha = \beta/(1-\beta)$ into above equation:

$$\underset{\mathcal{G}_s}{\arg\max}I(\mathcal{G}_s;\mathbf{Y}) + \alpha\left[I(\mathcal{G}_s;\mathcal{G}^*) - I(\mathcal{G}_s;\mathcal{G})\right].\tag{12}$$

Taking $\mathcal{G}\odot\mathbf{M}$ as the substitution of $\mathcal{G}_s$, the new form of GIB for explainability can be derived as:

$$\underset{\mathbf{M}}{\arg\max}I(\mathcal{G}\odot\mathbf{M};\mathbf{Y}) + \alpha\left[I(\mathcal{G}\odot\mathbf{M};\mathcal{G}^*) - I(\mathcal{G}\odot\mathbf{M};\mathcal{G})\right],\tag{13}$$

which is the objective of GIBE.

### A.2 PROOF OF THE CONVERGENCE OF GIBE

**Theorem A.1.** For any $\alpha \in (0, +\infty)$ we have: $\mathcal{G} \odot \mathbf{M} = \mathcal{G}^*$ maximizes the objective of GIBE: $I(\mathcal{G} \odot \mathbf{M}; \mathbf{Y}) + \alpha \left[ I(\mathcal{G} \odot \mathbf{M}; \mathcal{G}^*) - I(\mathcal{G} \odot \mathbf{M}; \mathcal{G}) \right]$.

*Proof.* Consider the following derivation,

$$
\begin{aligned}
& I(\mathcal{G} \odot \mathbf{M}; \mathbf{Y}) + \alpha[I(\mathcal{G} \odot \mathbf{M}; \mathcal{G}^*) - I(\mathcal{G} \odot \mathbf{M}; \mathcal{G})] \\
=& I(\mathbf{Y}; \mathcal{G} \odot \mathbf{M}, \mathcal{G}^*) - I(\mathcal{G}^*; \mathbf{Y} \mid \mathcal{G} \odot \mathbf{M}) + \alpha[I(\mathcal{G} \odot \mathbf{M}; \mathcal{G}^*) - I(\mathcal{G} \odot \mathbf{M}; \mathcal{G})] \\
=& I(\mathbf{Y}; \mathcal{G} \odot \mathbf{M}, \mathcal{G}^*) - I(\mathcal{G}^*; \mathbf{Y} \mid \mathcal{G} \odot \mathbf{M}) + \alpha[I(\mathcal{G}^*; \mathbf{Y}, \mathcal{G} \odot \mathbf{M}) - I(\mathcal{G}^*; \mathbf{Y} \mid \mathcal{G} \odot \mathbf{M}) \\
& + I(\mathcal{G}^*; \mathbf{Y} \mid \mathcal{G} \odot \mathbf{M}) - I(\mathcal{G}^*; \mathcal{G} \odot \mathbf{M}, \mathbf{Y})] \\
=& (1 + \alpha)I(\mathbf{Y}; \mathcal{G}^*) - (1 + \alpha)I(\mathcal{G}^*; \mathbf{Y} \mid \mathcal{G} \odot \mathbf{M}) + \alpha I(\mathcal{G}^*; \mathbf{Y} \mid \mathcal{G} \odot \mathbf{M}) - \alpha I(\mathcal{G}^*; \mathcal{G} \odot \mathbf{M}, \mathbf{Y}) \\
=& (1 + \alpha)I(\mathbf{Y}; \mathcal{G}^*) - I(\mathcal{G}^*; \mathbf{Y} \mid \mathcal{G} \odot \mathbf{M}) - 2I(\mathcal{G}^*; \mathcal{G} \odot \mathbf{M}, \mathbf{Y}) \\
=& (1 + \alpha)I(\mathbf{Y}; \mathcal{G}^*) - I(\mathcal{G}^*; \mathbf{Y} \mid \mathcal{G} \odot \mathbf{M}) - 2I(\mathbf{Y}; \mathcal{G}^*) - \alpha I(\mathcal{G}^*; \mathcal{G} \odot \mathbf{M} \mid \mathbf{Y}) \\
=& I(\mathbf{Y}; \mathcal{G}^*) - I(\mathcal{G}^*; \mathbf{Y} \mid \mathcal{G} \odot \mathbf{M}) - \alpha I(\mathcal{G}^*; \mathcal{G} \odot \mathbf{M} \mid \mathbf{Y}).
\end{aligned}
$$
$$(14)$$

Since $I(\mathbf{Y}; \mathcal{G}^*)$ is a constant, for any $\alpha \in (0, +\infty)$, $\mathcal{G} \odot \mathbf{M}$ that maximizes the objective of GIBE also minimizes $I(\mathcal{G}^*; \mathbf{Y} \mid \mathcal{G} \odot \mathbf{M}) + \alpha I(\mathcal{G}^*; \mathcal{G} \odot \mathbf{M} \mid \mathbf{Y})$. As $I(\mathcal{G}^*; \mathbf{Y} \mid \mathcal{G} \odot \mathbf{M}) > 0$ and $I(\mathcal{G}^*; \mathcal{G} \odot \mathbf{M} \mid \mathbf{Y}) > 0$, so the lower bound of $I(\mathcal{G}^*; \mathbf{Y} \mid \mathcal{G} \odot \mathbf{M}) + \alpha I(\mathcal{G}^*; \mathcal{G} \odot \mathbf{M} \mid \mathbf{Y})$ is 0.

We have $\mathcal{G} \odot \mathbf{M} = \mathcal{G}^*$ can make $I(\mathcal{G}^*; \mathbf{Y} \mid \mathcal{G} \odot \mathbf{M}) + \alpha I(\mathcal{G}^*; \mathcal{G} \odot \mathbf{M} \mid \mathbf{Y}) = 0$. This is because (1) $\mathbf{Y} = f(\mathcal{G}^*) + \epsilon$ where $\epsilon$ is independent of $\mathcal{G}$ so that $I(\mathcal{G}^*; \mathbf{Y} \mid \mathcal{G} \odot \mathbf{M}) = 0$ and (2) $\mathcal{G}^* = f^{-1}(\mathbf{Y} - \epsilon)$ where $\epsilon$ is independent of $\mathcal{G}$ so that $I(\mathcal{G}^*; \mathcal{G} \odot \mathbf{M} \mid \mathbf{Y}) = 0$.

Therefore, $\mathcal{G} \odot \mathbf{M} = \mathcal{G}^*$ maximizes the objective of GIBE in Equation 2.

### A.3 ESTIMATION OF MUTUAL INFORMATION IN FEATURE ATTRIBUTION

According to the definition of mutual information:

$$
I(\mathcal{G} \odot \mathbf{M}; \mathbf{Y}) = \mathbb{E}_{\mathcal{G} \odot \mathbf{M}, Y} \left[ \log \frac{\mathbb{P}(\mathbf{Y} \mid \mathcal{G} \odot \mathbf{M})}{\mathbb{P}(\mathbf{Y})} \right]. \tag{15}
$$

Since $\mathbb{P}(\mathbf{Y} \mid \mathcal{G} \odot \mathbf{M})$ is intractable, we introduce a variational approximation $\mathbb{P}_\theta(\mathbf{Y} \mid \mathcal{G} \odot \mathbf{M})$ for it. Then, we can rewrite Equation 15 as:

$$
\begin{aligned}
I(\mathcal{G} \odot \mathbf{M}; \mathbf{Y}) =& \mathbb{E}_{\mathcal{G} \odot \mathbf{M}, \mathbf{Y}} \left[ \log \frac{\mathbb{P}_\theta(\mathbf{Y} \mid \mathcal{G} \odot \mathbf{M})}{\mathbb{P}(\mathbf{Y})} \right] \\
& + \mathbb{E}_{\mathcal{G} \odot \mathbf{M}} \left[ \mathrm{KL}\left( \mathbb{P}(\mathbf{Y} \mid \mathcal{G} \odot \mathbf{M}) \| \mathbb{P}_\theta(\mathbf{Y} \mid \mathcal{G} \odot \mathbf{M}) \right) \right] \\
=& \mathbb{E}_{\mathcal{G} \odot \mathbf{M}, \mathbf{Y}} \left[ \log \mathbb{P}_\theta(\mathbf{Y} \mid \mathcal{G} \odot \mathbf{M}) \right] + H(\mathbf{Y}) \\
& + \mathbb{E}_{\mathcal{G} \odot \mathbf{M}} \left[ \mathrm{KL}\left( \mathbb{P}(\mathbf{Y} \mid \mathcal{G} \odot \mathbf{M}) \| \mathbb{P}_\theta(\mathbf{Y} \mid \mathcal{G} \odot \mathbf{M}) \right) \right].
\end{aligned}
\tag{16}
$$

### A.4 VARIATIONAL BOUNDS FOR MUTUAL INFORMATION IN FEATURE SELECTION

We now focus on the lower bounds for mutual information in the phase of feature selection:

$$
I(\mathcal{G} \odot \mathbf{M}; \mathcal{G}^*) - I(\mathcal{G} \odot \mathbf{M}; \mathcal{G}). \tag{17}
$$

For the first term $I(\mathcal{G} \odot \mathbf{M}; \mathcal{G}^*)$, by definition:

$$
I(\mathcal{G} \odot \mathbf{M}; \mathcal{G}^*) = \mathbb{E}_{\mathcal{G} \odot \mathbf{M}, \mathcal{G}^*} \left[ \log \frac{\mathbb{P}(\mathcal{G} \odot \mathbf{M} \mid \mathcal{G}^*)}{\mathbb{P}(\mathcal{G} \odot \mathbf{M})} \right]. \tag{18}
$$

Since $\mathbb{P}(\mathcal{G} \odot \mathbf{M} \mid \mathcal{G}^*)$ is intractable, we introduce a variational approximation $\mathbb{P}_\phi(\mathcal{G} \odot \mathbf{M} \mid \mathcal{G}^*)$ for it. Then, we obtain a lower bound for Equation 18:

$$
\begin{aligned}
I(\mathcal{G} \odot \mathbf{M}; \mathcal{G}^*) =& \mathbb{E}_{\mathcal{G} \odot \mathbf{M}, \mathcal{G}^*} \left[ \log \frac{\mathbb{P}_\phi(\mathcal{G} \odot \mathbf{M} \mid \mathcal{G}^*)}{\mathbb{P}(\mathcal{G} \odot \mathbf{M})} \right] \\
& + \mathbb{E}_{\mathcal{G} \odot \mathbf{M}, \mathcal{G}^*} \left[ \mathrm{KL}\left( \mathbb{P}(\mathcal{G} \odot \mathbf{M} \mid \mathcal{G}^*) \| \mathbb{P}_\phi(\mathcal{G} \odot \mathbf{M} \mid \mathcal{G}^*) \right) \right] \\
\geq& \mathbb{E}_{\mathcal{G} \odot \mathbf{M}, \mathcal{G}^*} \left[ \log \frac{\mathbb{P}_\phi(\mathcal{G} \odot \mathbf{M} \mid \mathcal{G}^*)}{\mathbb{P}(\mathcal{G} \odot \mathbf{M})} \right] \\
=& \mathbb{E}_{\mathcal{G} \odot \mathbf{M}, \mathcal{G}^*} \left[ \log \mathbb{P}_\phi(\mathcal{G} \odot \mathbf{M} \mid \mathcal{G}^*) \right] + H(\mathcal{G} \odot \mathbf{M}).
\end{aligned}
\tag{19}
$$

Then for the second term $I(\mathcal{G} \odot \mathbf{M}; \mathcal{G})$ we have:

$$I(\mathcal{G} \odot \mathbf{M}; \mathcal{G}) = \mathbb{E}_{\mathcal{G} \odot \mathbf{M}, \mathcal{G}} \left[ \log \frac{\mathbb{P}(\mathcal{G} \odot \mathbf{M} \mid \mathcal{G})}{\mathbb{P}(\mathcal{G} \odot \mathbf{M})} \right]. \tag{20}$$

Since $\mathbb{P}(\mathcal{G} \odot \mathbf{M})$ is intractable, we introduce a variational approximation $\mathbb{Q}$ for it. Then, we obtain a upper bound for Equation 20:

$$
\begin{aligned}
I(\mathcal{G} \odot \mathbf{M}; \mathcal{G}) &= \mathbb{E}_{\mathcal{G} \odot \mathbf{M}, \mathcal{G}} \left[ \log \frac{\mathbb{P}(\mathcal{G} \odot \mathbf{M} \mid \mathcal{G})}{\mathbb{Q}(\mathcal{G} \odot \mathbf{M})} \right] \\
&\quad - \mathbb{E}_{\mathcal{G} \odot \mathbf{M}, \mathcal{G}} \left[ \mathrm{KL}\left( \mathbb{P}(\mathcal{G} \odot \mathbf{M}) \| \mathbb{Q}(\mathcal{G} \odot \mathbf{M}) \right) \right] \\
&\leq \mathbb{E}_{\mathcal{G} \odot \mathbf{M}, \mathcal{G}} \left[ \log \frac{\mathbb{P}(\mathcal{G} \odot \mathbf{M} \mid \mathcal{G})}{\mathbb{Q}(\mathcal{G} \odot \mathbf{M})} \right] \\
&= \mathbb{E}_{\mathcal{G} \odot \mathbf{M}, \mathcal{G}} \left[ \mathrm{KL}\left( \mathbb{P}(\mathcal{G} \odot \mathbf{M} \mid \mathcal{G}) \| \mathbb{Q}(\mathcal{G} \odot \mathbf{M}) \right) \right].
\end{aligned} \tag{21}
$$

Plugging in Equation 19 and Equation 21, we obtain a variational lower bound of Equation 17 as the objective of feature selection:

$$
\begin{aligned}
I(\mathcal{G} \odot \mathbf{M}; \mathcal{G}^*) - I(\mathcal{G} \odot \mathbf{M}; \mathcal{G}) \geq &\mathbb{E}_{\mathcal{G} \odot \mathbf{M}, \mathcal{G}^*} \left[ \log \mathbb{P}_\phi(\mathcal{G} \odot \mathbf{M} \mid \mathcal{G}^*) \right] + H(\mathcal{G} \odot \mathbf{M}) \\
&- \mathbb{E}_{\mathcal{G} \odot \mathbf{M}, \mathcal{G}} \left[ \mathrm{KL}\left( \mathbb{P}(\mathcal{G} \odot \mathbf{M} \mid \mathcal{G}) \| \mathbb{Q}(\mathcal{G} \odot \mathbf{M}) \right) \right].
\end{aligned} \tag{22}
$$

## B  EXPERIMENT SETTING

**Datasets and Target GNNs.** We use one synthetic dataset and two real datasets which are publicly accessible. Three popular GNN models are trained to perform graph classification. The statistics of datasets and the configurations of GNN models are summarized in table 3. Note that some benchmark datasets may not satisfy the assumption in Section 2, for further exploration, we still take them into consideration.

- **MUTAG** (Kazius et al., 2005; Riesen & Bunke, 2008) contains 4,337 molecule graphs categorized into two classes based on their mutagenic effect on the Gram-negative bacterium.
- **BA-3motifs** (Ying et al., 2019; Luo et al., 2020) contains 3,000 graphs attaching with one of three motif types: house, cycle, and grid, where Barabasi-Albert (BA) graphs are adopted as the base.
- **MNIST** (Monti et al., 2017; Deng, 2012) superpixel dataset converts 70,000 images into the graphs of superpixel adjacency, where every graph is labeled as one of ten digit classes.

- **GIN** (Xu et al., 2019) suppresses the popular GNN variants, such as Graph Convolutional Networks and GraphSAGE in terms of expressive power, as it generalizes the Weisfeiler Lehman graph isomorphism test and hence achieves maximum discriminative power under the neighborhood aggregation framework.
- **ASAP** (Ranjan et al., 2020) utilizes a self-attention network along with a modified GNN formulation to capture the importance of each node in a given graph, and learns a sparse soft cluster assignment for nodes at each layer to effectively pool the subgraphs to form the pooled graph.
- **Spline-based GNN** (Fey et al., 2018) adopts a novel convolution operator based on B-splines, which operates in the spatial domain and aggregates local features, applying a trainable continuous kernel function parametrized by trainable B-spline control values and allowing very fast training and inference computation.

**Evaluation Metrics.** It is of crucial importance to evaluate the explanations quantitatively since human evaluations are highly dependent on their subjective understanding. Prior studies have proposed some metrics to quantitatively assess the explanations(Yuan et al., 2020; Dwivedi et al., 2020), among which we select three commonly used metrics to evaluate our results. For clarity, we denote $\mathcal{G}_s^K$ as the explanatory subgraph by taking top-$K$ edges in $\mathcal{G}$, and $|\mathcal{G}|$ as the number of edges in graph $\mathcal{G}$.

- **Predictive Accuracy (ACC@$\eta$)** (Chen et al., 2018). This metric feeds the explanatory subgraph into the target model and measures the performance of the explanation by auditing how well it recovers the target prediction, where $\eta$ is the predefined sparsity. Formally, given the trained GNN models $f$, we have

$$\mathrm{ACC@}\eta = \mathbb{E}_{\mathcal{G}}[\mathbb{I}(f(\mathcal{G}), f(\mathcal{G}_s^{\lceil (1-\eta) \times |\mathcal{G}| \rceil}))], \tag{23}$$

Table 3: Statistics of the datasets and configurations of GNN models.

|  | MUTAG | BA3-motif | MNIST |
|---|---|---|---|
| Graphs# | 4,337 | 3,000 | 70,000 |
| Classes# | 2 | 3 | 10 |
| Avg.Nodes# | 30.32 | 31.44 | 66.87 |
| Avg.Edges# | 30.77 | 31.24 | 725.39 |
| Target GNNs | GIN | ASAP | Spline-based GNN |
| Layers# | 2 | 2 | 5 |
| Testing Accuracy | 0.977 | 0.993 | 0.946 |

where $\mathbb{I}(\cdot, \cdot)$ is the indicator function that takes value 1 when its two arguments are equal and takes value 0 otherwise. Moreover, we plot the curve of ACC over different sparsity $\eta \in (0, 0.1, ..., 0.9)$ on the test set and denote ACC-AUC as the area under the ACC curve. Note that ACC@$\eta$ and ACC-AUC do not rely on ground truth labels, and thus are suitable for all the datasets.

- **Precision**@$N$ (Ying et al., 2019). This metric measures the consistency between the explanatory subgraph $\mathcal{G}_s$ and the ground-truth subgraph $\mathcal{G}^*$. Concretely, the edges within $\mathcal{G}^*$ are positive in $\mathcal{G}$, while the remains are negative. In this case, precision can be adopted as the evaluation protocol. More formally,

$$\text{Precision@}N = \mathbb{E}_{\mathcal{G}} \left[ \frac{|\mathcal{G}_s^N \bigcap \mathcal{G}^*|}{|\mathcal{G}^*|} \right] \tag{24}$$

- **Fidelity**@$p$ (Yuan et al., 2020). The Fidelity metric studies the prediction change by removing important input features identified by explanation methods. Formally,

$$\text{Fidelity @}p = \mathbb{E}_{\mathcal{G}} \left[ f\left(\mathcal{G}\right)_{\mathbf{Y}} - f\left(\mathcal{G} \setminus \mathcal{G}_s^{\lceil p \times |\mathcal{G}| \rceil}\right)_{\mathbf{Y}} \right] \tag{25}$$

**Baselines.** We leverage the state-of-the-art methods to verify proposed propositions and optimization schemes, covering the followings:

- **GNNExplainer** (Ying et al., 2019) directly learns an adjacency matrix mask through maximizing the mutual information between a GNN's prediction and distribution of possible subgraph structures, which is performed via multiplying the mask to the input features.
- **PGExplainer** (Luo et al., 2020) adopts a deep neural network to parameterize the generation process of explanations, which makes it a natural approach to explaining multiple instances collectively. It can also provide global explanations for a certain class.
- **GraphMask** (Schlichtkrull et al., 2021) learns a simple classifier that, for every edge in every layer, predicts if that edge can be dropped, in a fully differentiable fashion. Then by dropping edges without deteriorating the performance of the model, the remaining edges naturally form an explanation for the model's prediction.
- **CF-GNNExplainer** (Lucic et al., 2022) focuses on the counterfactual explanations by figuring out the minimal perturbation to the input (graph) data such that the prediction changes. By instantiating the perturbation with only edge deletions, they find out the edges which are crucial for the original predictions.
- **Refine** (Wang et al., 2021) develops a explainer that can generate multi-grained explanations by exploiting the pre-training and fine-tuning idea. Specifically, the pre-training phase exhibits global explanations with the prototypical patterns, and the fine-tuning phase further adapts the global explanations in the local context with high fidelity.
- **GSAT** (Miao et al., 2022) leverages stochastic attention to block the information from the task-irrelevant graph components while learning stochasticity-reduced attention to select the task-relevant subgraphs for interpretation. Though it's naturally an inherently interpretable method, it also works in a post-hoc way through a fine-tuning fashion. We adopt this post-hoc working mode as one baseline.

**Training Optimization and Early Stopping.** All experiments are done on a single Tesla V100 SXM2 GPU (32 GB). During training, we use Adam (Kingma & Ba, 2015) optimizer. The maximum number of epochs is 200 for all datasets. We use Stochastic Gradient Descent (SGD) for the optimization of all GNNs models. The initial learning rate is set to: $10^{-3}$ for BA3-motif , $10^{-2}$

for MNIST, and $10^{-3}$ for MUTAG. Also, we exhibit early stopping to avoid overfitting the training dataset. If the model's performance on the validation dataset is without improvement (*i.e.*, validation accuracy begins to decrease) for five epochs, we stop the training process to prevent increased generalization error.

**Hyperparameter Settings.** The most crucial hyperparameter in this work is the coefficients of regularization. For MUTAG: the initial coefficient of sparsity constraints is $0.05$ and it grows at a rate of $50\%$ each epoch; the coefficient of discrete constraints is $0.5$ and it grows at a rate of $50\%$ each epoch; For BA3, these coefficients are set to $\{0.04, 50\%, 0.4, 50\%\}$; For MNIST: these coefficients are set to $\{5 \times 10^{-5}, 100\%, 5 \times 10^{-4}, 50\%\}$.

For other hyperparameters in baseline methods, we adopt a grid search for the optimal parameters using the validation datasets. To be more specific, the learning rate of Adam is tuned in $\{10^{-3}, 10^{-2}, 10^{-1}\}$, and the weight decay is searched in $\{10^{-5}, 10^{-4}, 10^{-3}\}$. Other model-specific hyperparameters are set as follows: For PGExplainer, the temperature for reparameterization is $0.1$; For Refine, the temperature hyperparameter $\beta$ is 1 and the trade-off hyperparameter $\gamma$ is 5; For GSAT, the parameter of Bernoulli distribution $r$ is fixed as $0.6$.

## C  QUALITATIVE EVALUATION OF SRS

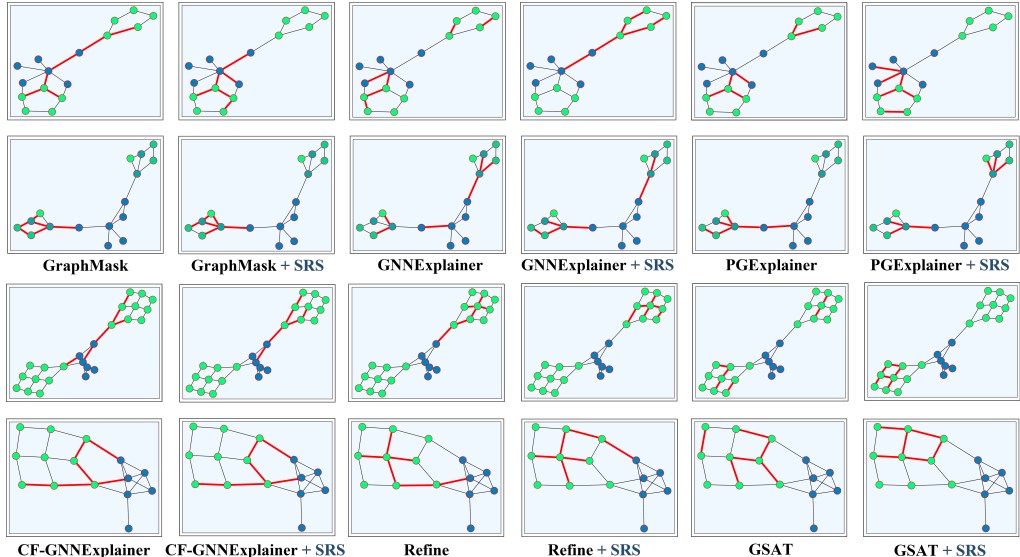

Figure 6: Selected explanations in BA3-motif, where the top-6 of directed edges are highlighted by red lines. The ground-truth nodes are highlighted in green while the turbulence nodes are distinguished in blue. Best viewed in color.

To have visual inspections on the explanatory subgraphs generated by different explainers and the effectiveness of our proposed scheme SRS, we randomly choose graph instances from the synthetic dataset BA3-motif and present them in Figure 6. For each baseline explainer, we highlight the edges which have the top-$K$ importance scores by red lines, where $K = 6$. The ground-truth nodes are highlighted in green while the turbulence nodes *w.r.t.* nodes in BA-motif are distinguished in blue. According to Figure 6 we can observe that:

- Some blue edges which is not belonging to the ground-truth motifs are select by the baseline explanation methods. On the contrary, current explainers enhanced by our proposed scheme SRS inherit the ground-truth edges extracted by the original methods, and eliminate the edges which belong to the Barabasi-Albert (BA) graphs.
- Current explainers enhanced by SRS take count of both accuracy and completeness. That is, SRS mainly focuses on the edges belonging to one completed motif while the graph contains more than one ground-truth motifs. However, baseline explainers are often distracted by multiple ground-truth motifs.

- For blue nodes in Barabasi-Albert (BA) graphs, there are several more intrusive nodes which are connected to ground-truth motifs. These nodes will cause higher interference to the generation of explanatory subgraphs. Fortunately, SRS framework can avoid these traps by extracting less turbulence nodes. This phenomenon demonstrates the robustness of our proposed SRS.

