# OpenReview forum: "On Regularization for Explaining Graph Neural Networks: An Information Theory Perspective"
_ICLR.cc/2023/Conference — Submitted to ICLR 2023_

### Official Review · Reviewer_SjeN · 2022-10-21

**Confidence:** 2
**Correctness:** 3
**Technical Novelty And Significance:** 3
**Empirical Novelty And Significance:** 3
**Recommendation:** 6

**Clarity, Quality, Novelty And Reproducibility:**

The paper is novel to the extent that GIBE is an extension of GIB for explanability of GNNs. However, the manuscript could be significantly improved when it comes to clarity, as for example, the follow up to the proposed Propositions are not as clearly discussed by the authors (see e.g., my question on SRS).

**Strength And Weaknesses:**

Strength
======

- The paper introduces GIBE, which is similar to GIB but for GNNs explainability.  To my understanding, this is novel.
- The authors do a good job at discussing the related literature and putting their work into perspective.

Weakness
========

- Figures 4 and 5 could definitely be improved by using different line formats and markers to facilitate the task of distinguishing the results.
- Overall, the proposed methodology in the paper is not easy to digest. In particular, I'm confused about the exact implementation/derivation of the proposed SRS method. Could you elaborate more on the rationale behind it? Is it just a grid search on $\eta$ and then scaling $\mathbf{K}$ by the best $\eta$?

Questions
========

- What is the role of $\alpha$ in Eq (2) given that its optimal solution is independent of it?

**Summary Of The Paper:**

This paper investigates the role of regularization in the explainability of GNNs. They find that regularization pursues a balance between feature attribution and selection as well as that optimal regularization is related to the sparsity of the explanations.

**Summary Of The Review:**

Overall it's a promising paper, but the clarity/justification of the proposed methodology could be improved.

---

> ### Author Response · Authors · 2022-11-11
> **Response to reviewer SjeN.**
>
> Thank you for the valuable feedback! We have modified our representation in the updated version under the guidance of your comments. We would appreciate it if you could take another look!
>
> Below, we give point-by-point responses to your comments and describe the revisions we made to address them. We would be happy to add additional clarifications and revisions to the paper to address any additional recommendations from you.
>
> >***Q1:  Figures 4 and 5 could definitely be improved by using different line formats…***
>
> Thanks for your suggestion! We have diversified the line formats in Figures 4 and 5.
>
> As for the line markers, since we show the variance by the length of markers, we have tested the presentation of several markers, and found that the uniform marker could better present the relative magnitude of the variance. Thus we haven't made changes to the markers. We especially hope to hear your understanding.
>
> What’s more, we have polished our representation and emphasized the meaning of the length of the marker. Meanwhile, if you consider the diverse markers are necessary, we are more than happy to address this further!
>
> >***Q2: …the proposed SRS method…Could you elaborate more on the rationale behind it?***
>
> Sorry for that we don’t make it clear. According to *Proposition 2*, the optimal coefficients of regularization are proportional to the sparsity. Based on this, SRS first seeks for the optimal coefficients of regularization under a certain sparsity by grid search, then for other sparsity, SRS changes the coefficients according to the fluctuation of sparsity .
>
> What’s more, as per your suggestion, we have polished our representation in Section 4.1 and Section 5. Please see if the description in the reversion clarifies your concerns.
>
> >***Q3: What is the role of $\alpha$ in Eq. (2) given that its optimal solution is independent of it?***
>
> $\alpha$ is the trade-off parameter in Eq. (2) to balance the phases of feature attribution and selection.
> Specifically,
>
> + if $\rm{M}$ in Eq. (2) is equal to its optimal solution, the second term of Eq. (2)  will be equal to zero. In this case, $\alpha$ serves no functional purpose.
>
> + However, most of current explainers do not have the capacity to find the precise optimal solution. In other words, the second term of Eq. (2) isn’t equal to zero. In this case, the solution of Eq. (2) boils down to a tradeoff between its two terms, where $\alpha$ acts as a supportive role to keep the balance.
>
> Thank you again for the valuable feedback! Please let us know if you have any other comments/questions, we are more than happy to address them further!
>
> Best,
>
> Authors

---

> > ### Comment · Reviewer_SjeN · 2022-11-17
> > **Additional Comments**
> >
> > Some comments on the writing style:
> >
> > I personally wouldn't use the words "What's more" in the manuscript as it sounds too informal for academic writing. You could perfectly replace those for a more formal version such as "Additionally" or "Furthermore".
> >
> > Like a said in my original review, the methodology of this paper doesn't feel easy digest, even after major changes done by the authors. For example, in your response when you say "SRS changes the coefficients according to the fluctuation of sparsity", what do you mean by "fluctuation of sparsity"?

---

> > > ### Author Response · Authors · 2022-11-17
> > > **Response to reviewer SjeN**
> > >
> > > Dear reviewer:
> > >
> > > I especially appreciate your response! Below, we give responses to your comments and describe the revisions we made to address them. We are more than happy to add additional clarifications and revise the paper to address any additional recommendations from you!
> > >
> > > >***Q1: …You could perfectly replace those for a more formal version such as "Additionally" or "Furthermore".***
> > >
> > > Thanks for your valuable suggestions! We have polished our representation in the updated version, and we will take note of that in the future.
> > >
> > > >***Q2: …SRS changes the coefficients according to the fluctuation of sparsity", what do you mean by "fluctuation of sparsity"?***
> > >
> > > Thanks for your concern. Please see the fourth paragraph of Section 5.2, where we elaborate the process of SRS.
> > >
> > > Additionally, we have polished our representation in Section 4.1 for easy of understanding. Specifically, SRS first seeks for the optimal regularization coefficients $\rm{K}_i$ under the certain sparsity $\eta_i$ by grid search, then for other sparsity $\eta_j$, SRS increases or decreases coefficients proportionally to the variation of sparsity. Formally, $\rm{K}_j=(\eta_j / \eta_i) \rm{K}_i$. In a nutshell, SRS is a simple but effective optimal framework for GNNs explanation methods derived from Proposition 2. We would appreciate it if you could take another look.
> > >
> > > Thank you again for the valuable feedback! Please let us know if you have any other comments/questions!
> > >
> > > Best,
> > >
> > > Authors

---

### Official Review · Reviewer_eWP9 · 2022-10-24

**Confidence:** 3
**Correctness:** 2
**Technical Novelty And Significance:** 2
**Empirical Novelty And Significance:** 2
**Recommendation:** 1

**Clarity, Quality, Novelty And Reproducibility:**

As commented above, the clarity of this paper needs to be largely improved. It is difficult to assess the technical soundness due to the lack of clarity. As far as I can tell, the analysis seems to be limited to a specific GNN and cannot be applied to the common post-hoc GNN explainers, which limits the significance of the contribution.

**Strength And Weaknesses:**

Strength: The topic of this paper, theoretical understanding of GNN explainers, is an important research direction.


Weaknesses:

1) The writing of this paper needs to be largely improved. Many essential concepts are not well-defined in the main paper or the appendix. For example, what are the distribution assumptions on the graph data? In section 3.2, "Treating target GNN f' as the proxy function ...", what is f'? What is the formal definition of "regularization" in GIBE? The technical content is almost unreadable due to missing definitions.


2) The paper claims to provide an understanding of the role of regularization for existing GNN explainers. However, the analysis is specific to a particular type of GNN (proposed by Miao et al. 2022) that has a certain interpretable mechanism. It is unclear how the analysis can be transferred to the more commonly used post-hoc explanation methods that can be applied to different types of GNNs.

3) Many typos. For example,

- right before section 3.2, "... of Equation 2 can be proved to equal to ...": to be equal to.
- Many ":" should be replaced by ",", e.g., "... employing Data Processing Inequality (DPI) along the Markov chain**:** ...".
- Proposition 2, "let K_i and K_j is the ...": let K_i and K_j be the ...



**Summary Of The Paper:**

This paper aims to investigate the role of regularization used in existing GNN explainers. The paper analyzes the interpretable GNN model proposed by Miao et al. 2022 by rewriting the training objective function and mapping it into two parts, the feature attribution objective and the feature selection objective. And the authors claim to derive some findings about the "regularization" in this rewritten objective.

**Summary Of The Review:**

Given the lack of clarity and the limited scope of analysis, I'm inclined to reject this paper as the problems are unlikely to be addressed during the author response period.

---

> ### Author Response · Authors · 2022-11-11
> **Response to reviewer eWP9.**
>
> Thank you for the valuable comments. As per your suggestion we have modified our representation in the updated version. We would especially appreciate it if you could take another look!
>
> Below, we give responses to your comments and describe the revisions we made to address them. We would be happy to add additional clarifications and revisions to the paper to address any additional comments from you.
>
> >***Q1.1: …what are the distribution assumptions on the graph data?...***
>
> Thanks for your concern. We have modified our representation in Section 2 and Section 4.3. Please see if the modification in the reversion clarifies your concerns. Specifically,
>
> + For the distribution of the input graph, we assume that the input graph is independent and identically distributed (IID) from the data distribution.
>
> + For the distribution of subgraph, we argue that there are distribution shifts between subgraph and the original graph in the data space. For example, there might be size OOD [1], homophily OOD [2] and degree OOD [3] in the data space.
>
> [1] Size-Invariant Graph Representations for Graph Classification Extrapolations. ICML 2021
>
> [2] EvenNet: Ignoring Odd-Hop Neighbors Improves Robustness of Graph Neural Networks. NIPS2022
>
> [3] Investigating and Mitigating Degree-Related Biases in Graph Convolutional Networks. CIKM 2020
>
> >***Q1.2: …Treating target GNN $f'$ as the proxy function ...", what is $f'$?.***
>
> Thanks for your concern. $f'$ is the classifier for input graph $\mathcal{G}$. For easy of understanding, we have polished our representation in terms of $f'$ in Section 2.
>
> >***Q1.3: …What is the formal definition of "regularization" in GIBE?...***
>
> Similarly with GIB, GIBE doesn’t contain the terms involving the definition of regularization directly. The bridge between GIBE and the regularization is the training process of $\rm{M}$. That is, regularization acts on GIBE by affecting the generation of $\rm{M}$ and further impacting the balance between the two terms in GIBE.
>
> >***Q2: …It is unclear how the analysis can be transferred to the more commonly used post-hoc explanation methods…***
>
> Thanks for your concern. **GIBE and our propositions are applicable to most of the explainable methods.** Actually, unifying current explainers into a universal paradigm is the main objective of our work. Below, we present reasons step by step.
>
> + Please see the second paragraph of Section 3.1, where we demonstrate that the paradigm of $\mathcal{G} \cdot  \rm{M}$ is appropriate for the mask-based, the attention-based and the perturbation-based post-hoc explanation methods [1,2,3,4,5,6,7], and so is GIBE.
>
> + In Section 3.2, 4.1 and 4.2, all the propositions are derived from GIBE and the theorems in information theory. Note that there are no additional conditions or assumptions involved. That is, GIBE and our propositions are applicable to most of the explainable methods contain but not limited to [Miao et al.(2022)](https://proceedings.mlr.press/v162/miao22a.html).
>
> + We have verified our analysis and propositions by extensive experiments across six explanation methods (*i.e.*, GNNExplainer [1], PGExplainer [2], GraphMask [3], CF-GNNExplainer [4], Refine [5] and GSAT [6]).
>
> Moreover, we are also encouraged that all the other reviewers (Reviewers eWP9 and SjeN) find the universality of our theoretical analyses. Specifically,
>
> + Reviewer SjeN: “ *The authors do a good job at discussing the related literature and putting their work into perspective.* ”
>
> + Reviewer eWP9: “ *I like the idea of unifying most existing methods with GIB.* ”
>
> Nevertheless, we have polished our representation and emphasized the universality of our analysis in Section 3 and 4. We would especially appreciate it if you could take another look!
>
> [1] GNNExplainer: Generating Explanations for Graph Neural Networks. NIPS2019
>
> [2] Parameterized Explainer for Graph Neural Network. NIPS2020
>
> [3] Interpreting Graph Neural Networks For NLP With Differentiable Edge Masking. ICLR2021
>
> [4] CF-GNNExplainer: Counterfactual Explanations for Graph Neural Networks. AISTATS 2022
>
> [5] Towards Multi-Grained Explainability for Graph Neural Networks. NIPS2021
>
> [6] Interpretable and Generalizable Graph Learning via Stochastic Attention Mechanism. KDD2022
>
> [7] Robust Counterfactual Explanations on Graph Neural Networks. NIPS2021
>
> >***Q3: …Many typos. For example…***
>
> Thanks for your concern. As per your suggestions, we have modified our representation in the updated version. Please see if the modification in the reversion clarifies your concerns.
>
> Thank you again for the valuable feedback! Please let us know if you have any other comments/questions, we are more than happy to address them further!
>
> Best,
>
> Authors

---

> > ### Comment · Reviewer_eWP9 · 2022-11-17
> > **Thanks for the response**
> >
> > I appreciate the responses by the authors.
> >
> > However, the points I raised in Q1 are just some examples and the technical content is still fairly handwavy here and there.
> >
> > For example, the authors state that "some regularization in existing explainers lacks concrete theoretical support". What exactly is the "regularization" in existing explainers referred to here? Why there cannot be an explicit definition of that?
> >
> > In another example, "Y = f(G^∗) + ϵ for some deterministic invertible function f". Could you give a concrete example of such an "invertible" function with a graph as input?

---

> > > ### Comment · Reviewer_eWP9 · 2022-11-17
> > > **Additional comments**
> > >
> > > This paper may have some significant contributions. But the lack of clarity in the current writing largely hinders the verification of its correctness. And I believe the lack of clarity will also cause significant problems for its future readers, even if the major conclusions are correct. The clarity problem is also captured by Reviewer SjeN.
> > >
> > > My overall evaluation is mainly based on this significant clarity problem and I deem it not ready for publication in its current form or after minor revisions.

---

> > > > ### Author Response · Authors · 2022-11-17
> > > > **Response to reviewer eWP9**
> > > >
> > > > Dear reviewer:
> > > >
> > > > We especially appreciate your response. Following your previous suggestions for the presentation and logic, we have modified the manuscript to address all the concerns in the updated version. Below, we give responses to your new comments and describe the revisions we made to address them.
> > > >
> > > > However, we respectfully think that you mainly focus on presentation details within individual sentences. **We do not think the presentation details should be the main reason for the unusual rating of strong reject**. As for your claim of "*handwavy*", since our work is the first attempt to unify the current explainers from the perspective of feature attribution and selection, our theoretical content focuses on elaborately analyzing the common components in explainers, which is not "*handwavy*".
> > > >
> > > >
> > > > > *Q1: What exactly is the "regularization" in existing explainers referred to here?*
> > > >
> > > > Sorry for that we don’t make it clear. Formally, "regularization" in existing explainers refers to the constraint term on the property of explanatory subgraphs, which is introduced to guide the process of feature selection. Specifically, except the description in the second paragraph of Section 1, please see the third paragraph of Section 2, where we denote the regularization by specific terms in existing explainers like $l_1$ norm, connectivity constraints, and information constraints.
> > > >
> > > > Moreover, we have modified the representation in the first paragraph of Section 1 to address this concern. We would especially appreciate it if you could take another look!
> > > >
> > > > > *Q2: Could you give a concrete example of such an "invertible" function with a graph as input?*
> > > >
> > > > Thanks for your concern. We wonder if the following examples could address your concern:
> > > >
> > > > BA3-motif is one of the benchmark datasets for evaluating explanation methods. It contains 3,000 graphs attaching with motif types: *house*, *cycle*, or *grid*. On the one hand, if a graph contains the *house* motif (*i.e.*, $\mathcal{G}^\ast$), its label should be "House" (*i.e.*, $\rm{Y}$). On the other hand, if a graph belongs to the "House" class, it should contain the *house* motif. In this case, $\rm{Y}$ and $\mathcal{G}^\ast$ are bridged by an invertible function.
> > > >
> > > > Thank you again for the valuable feedback. Please let us know if you have any other comments/questions, we are more than happy to address them further!
> > > >
> > > > Best,
> > > >
> > > > Authors

---

> > > > > ### Comment · Reviewer_eWP9 · 2022-11-17
> > > > > **Thanks for the follow up**
> > > > >
> > > > > Thank the authors for the follow-up.
> > > > >
> > > > > The BA3-motif example provided by the authors is indeed a valid example of an invertible function.
> > > > >
> > > > > I would like to use this example to illustrate why I deem that the technical content in this submission is handwavy.
> > > > >
> > > > > First, there is an implicit data assumption behind the sentence "More formally, Y is determined by G^∗ in the sense that Y = f(G^∗) + ϵ for some deterministic invertible function f (Miao et al., 2022), where randomness ϵ is independent from G".
> > > > >
> > > > > To make it rigorous and improve the clarity, one should better explicitly state the assumption with something like the following statement:
> > > > >
> > > > > - We assume that there exists an invertible function $f: \mathcal{G}^* \rightarrow \mathcal{Y}$, where $\mathcal{G}^*$ is a subset of all possible graphs and $\mathcal{Y}$ is the label space. We further assume that for each data sample $G$, there exists a $G^* \in \mathcal{G}^*$, which is a subgraph of $G$, such that the label $Y$ for this sample is determined by $Y = f(G^*) + \epsilon$, for some $\epsilon$ as noise independent of the sample $G$.
> > > > >
> > > > > In the original writing, the domain and codomain of $f$ are not specified. And it is confusing to talk about invertible functions without knowing the domain and codomain. The example statement I gave above is still a bit problematic as the form of the noise $\epsilon$ is still unclear. And maybe the codomain of $f$ should not be the whole label space depending on the form of the noise.
> > > > >
> > > > > It is only with this level of clarity, we can start to examine the technical soundness in a relatively easy way. And for this specific part, apparently the assumption is overly strong. It is difficult to imagine such data assumptions would hold on moderately more realistic datasets, such as the MNIST and MUTAG datasets used in this paper.

---

> > > > > > ### Author Response · Authors · 2022-11-17
> > > > > > **Response to reviewer eWP9**
> > > > > >
> > > > > > Dear reviewer:
> > > > > >
> > > > > > We especially appreciate your super quick reply and thoughtful comments!
> > > > > >
> > > > > > We agree with that there should be explicit statement about the assumption following [Miao et al.(2022)](https://proceedings.mlr.press/v162/miao22a.html). We have added this part to Section 2 in the updated version under the guidance of your comments. Specifically, some benchmark datasets (*e.g.*, BA3-motif) of GNNs explainability indeed satisfy the above assumption, and whether some other datasets (*e.g.*, MUTAG) satisfy remains to be explored. We have also added demonstration to the experiment part in Appendix B. **However, we think this point should not erase the *significant contributions* (*which fortunately win your recognition in your additional comments*) of our work.** One the one hand, unifying explainers from the perspective of feature attribution and selection is the GNNs explainability sorely needs, and to explore the existence of ground-truth structure is exactly one of the targets of the current explainers. On the other hand, under the above assumption, we have verified our theoretical analysis and propositions across the datasets like BA3-motif and the datasets like MUTAG. All the results showed remarkably consistency. These results not only experimentally verify the rationality of our analysis, but also somehow help the readers further explore the properties of the datasets. We believe they are beneficial for the development of the community. We especially look forward to your understanding！
> > > > > >
> > > > > > Thank you again for the valuable feedback. Please let us know if you have any other comments/questions, we are more than happy to address them further!
> > > > > >
> > > > > > Best,
> > > > > >
> > > > > > Authors

---

> > > > > > > ### Comment · Reviewer_eWP9 · 2022-11-17
> > > > > > > **I did NOT acknowledge the significance of the contribution.**
> > > > > > >
> > > > > > > Thanks the authors for updating the submission, although the updated version is still problematic as it suggests the domain of f is all possible subgraphs instead of a certain subset of subgraphs.
> > > > > > >
> > > > > > > Let me make it clear that my original overall evaluation remains the same after all the discussions. I'd like to highlight two main reasons for my recommendation:
> > > > > > >
> > > > > > > - First, the significance of the contribution of this work relies on the technical soundness of the analysis, which is difficult to assess due to the lack of clarity **throughout the paper**. So I did NOT acknowledge the significance of the contribution in my previous posts.
> > > > > > > - Second, even if the major conclusions of this paper hold, the clarity problem is still a **major concern** for this paper to be published.
> > > > > > >
> > > > > > > As a minor point, the overly strong data assumptions as we discussed above already cast doubt on the technical soundness of this work.

---

> > > > > > > > ### Comment · Area_Chair_1f9F · 2022-11-29
> > > > > > > > **Thank you**
> > > > > > > >
> > > > > > > > Dear reviewer,
> > > > > > > >
> > > > > > > > Thanks for clarifying this and providing concrete examples.
> > > > > > > >
> > > > > > > > AC

---

> > > > ### Comment · Area_Chair_1f9F · 2022-11-23
> > > > **Please be more concrete with your criticism**
> > > >
> > > > Dear reviewer,
> > > >
> > > > You have to be more precise and concrete with your criticism about the lack of clarity. Where in the paper and what specifically lacks clarity? If you allow me, your review lacks clarity, because you don't describe what is missing in your opinion. If you don't update your review and/or provide additional context, I won't be able to incorporate your score in the meta-review.
> > > >
> > > > Thank you for your understanding.
> > > >
> > > > AC

---

> > > > > ### Comment · Reviewer_eWP9 · 2022-11-23
> > > > > **Response to AC**
> > > > >
> > > > > Dear AC,
> > > > >
> > > > > Thank you for the suggestion. The technical content of this paper is in general very handwavy. Since the major contribution of this paper is claimed to be theoretical, this type of lack of clarity is unacceptable.
> > > > >
> > > > > While the whole technical content needs to be improved here and there which I could not enumerate, I did have mentioned some examples in both the original review as well as the many-round follow-up discussion threads. Nevertheless, I'm summarizing them again below:
> > > > >
> > > > > - Mentioned in my original review:
> > > > >     - A main subject of interest in this paper is the "regularization" in GNN explainers. However, the "regularization" is never formally defined.
> > > > >     - Originally, some notations were used without definition (e.g., f'). Now some were fixed in the revision, but the definitions of some are still handwavy.
> > > > >     - The graph data distribution was not specified.
> > > > >
> > > > > - Mentioned in discussion:
> > > > >     - How f could be an invertible function mapping from (sub)graphs to labels and what are the data assumptions introduced?
> > > > >     - For the above question, I even gave an example of rewriting to make the data assumptions explicit (in my last-second post to the authors). The authors leveraged that into the revised paper. However, the revised version is still not crystally clear about the assumptions.
> > > > >
> > > > > In addition to the clarity problem. I also raised the question that the data assumptions introduced by the invertible function cannot hold for moderately realistic datasets.
> > > > >
> > > > > If the AC allows me, I encourage the AC to also read the discussion thread as well as to skim the paper, since this paper has diverging scores (although the lack of clarity of the technical content is also pointed out by Reviewer SejN).

---

### Official Review · Reviewer_HoXU · 2022-10-25

**Confidence:** 4
**Correctness:** 4
**Technical Novelty And Significance:** 3
**Empirical Novelty And Significance:** 3
**Recommendation:** 6

**Clarity, Quality, Novelty And Reproducibility:**

The writing is overall clear. The authors make use of the figures to help illustrate the key ideas. In terms of technical novelty, the authors could better highlight their own contributions. Some contents in the current version are overlapped with existing works.

**Strength And Weaknesses:**

Strengths:
- I like the idea of unifying most existing methods with GIB.
- There are discussions for the relationships between GIBE and sparsity, stochastic attention mechanism and OOD.
- Experimental results demonstrate the efficacy of the proposed techniques.

Concerns:
- In section 1, is it possible to better explain why the rank is better than L1 norm? A low-rank matrix is not necessarily sparse, but L1 norm can help control the size of the graph (#edges).
- Regarding GIB, how do Eq. (1) ensure the sparsity of subgraph? And how do we define the subgraph are sparse enough? Moreover, for some motif structure, it might not be sparse but is very important in determining specific properties of the graph (e.g., the hydroxide group -OH in some molecular graphs). In this case, should we consider such motifs if we want the explanatory subgraph to be sparse?
- Current definition 2 is not self-contained. It is better to explain what $G*$ is in the definition.
- A similar definition of GIBE is presented in Theorem 4.1 in (Miao et al. 2022). It would be great to discuss the similarity and difference with that formulation.
- I think some contents in section 3 (or maybe previous sections) needs adjustment. This is actually a key section to show that most existing methods fall into this GIB-type formulation and to explain the regularization between attribution and selection. This key connection is somehow not clear in previous sections. I get the key idea of this paper until I finished reading section 3.
- It is better not to use gap1 and gap2 in the main body, as this is specific to Figure 1 and will confuse readers.
- I didn't fully understand the purpose of Q in Eq. (4). Analytically, why do the introduction of variational approximation Q make it constant while original marginal distribution P is not?
- Most contents in section 3, section 4.1 and section 4.2 are already studied in (Miao et al. 2022). It is better to spend more efforts in highlighting the contributions by the authors.
- In addition to quantitative analysis on the proposed method, I am more interested in qualitative analysis like case studies. This is related to one last long-standing question of myself: We say explainability helps understand the behavior of black-box models. But why can we trust the explanations generated by another black-box model(s)? How should we trust this black-box explanation generator?

**Summary Of The Paper:**

This paper studies GNNs' explainability from the perspective of graph information bottleneck (GIB). The authors first provide justification on why most existing methods are essentially studying the tradeoff between attribution and selection by modeling them as GIB questions. Based on this observation, the authors discuss the relationship between regularizing the selection in sparsity, stochastic attention and out-of-distribution (OOD) issue, and propose the SRS and ORS scheme. Experimental results demonstrate that the proposed SRS and ORS improves the fidelity and accuracy of the proposed method.

**Summary Of The Review:**

The paper is interesting overall. It tries to explain the inherent regularization between attribution and selection for most existing works with the GIBE formulation. With the GIBE formulation, the authors discuss its relationships to sparsity, stochasticity and OOD. The technical contributions could be better highlighted in the literature. And some qualitative analysis would be helpful in understanding the superiority of the proposed method in real world scenarios.

---

> ### Author Response · Authors · 2022-11-11
> **Response to reviewer HoXU.**
>
> >***Q5: some contents in Section 3…maybe previous sections needs adjustment.***
>
> We agree. We have modified our representation in Section 1 to emphasize the important connection function of Section 3. Please see if the modification clarifies your concern.
>
>  >***Q6: It is better not to use gap1 and gap2 in the main body…***
>
> Thanks for your suggestion. We have replaced them in the updated version.
>
>  >***Q7: I didn't fully understand the purpose of $\mathbb{Q}$ in Eq. (4)…***
>
> Sorry for that we don’t make it clear.  Since $P(\mathcal{G} \odot \rm M)$ is intractable, we leverage distribution $\mathbb{Q}$ to approximate it for deriving a tractable lower boundary of the Eq. (4). We believe the form of Eq. (4) containing $\mathbb{Q}$ can provide active reference for the followers. For example, $\mathbb{Q}$ can be instantiated by the trained deep learning models to get a more precise lower boundary [1,2].
>
> [1] Graph Information Bottleneck for Subgraph Recognition. ICLR 2021
>
> [2] Interpretable and Generalizable Graph Learning via Stochastic Attention Mechanism. ICML2022
>
>  >***Q8: …It is better to…highlighting the contributions in section 3, section 4.1 and section 4.2.***
>
> Thanks for your suggestion. We have highlighted our contributions in Section 3 and 4. Specifically, Section 3 and 4 mainly focus on unifying the common paradigm of existing methods and rethinking the role of regularization in them, including but not limited to the theory in [Miao et al.(2022)](https://proceedings.mlr.press/v162/miao22a.html).
>
> Below we summarize the detailed differences between the theoretical parts of our work and Miao et al.(2022) in the following table:
>
> |                                  |             $~~~~~~~~~~~~~~~~~~~~~~~~~~~~~~~~~$           Our Work                     |     $~~~~~~~~~~~~~~~~~~~~~~~~$   [Miao et al.(2022)](https://proceedings.mlr.press/v162/miao22a.html)              |
> |-------------------------------:|:------------------------------------------------------:|:---------------------------------------------:|
> |                        Motivation |             Unifying explainers into a common paradigm           |   Promote explainability and generalization   |
> |                    Perspective | Feature attribution and selection & Information theory |               Information theory              |
> |              Theoretical Basis |                          GIBE                          |                      GIB                      |
> |    Rationale of Regularization |                      Keeping balance                      |          Constraining mutual information         |
> |        Scope of Regularization |               Most of current constraints, including $l_1$ norm and $l_2$ norm et, al      |             Information constraint            |
> |     Rationale of Stochasticity |                 Implicit regularization                |              Attention mechanism              |
> | Instantiation of Stochasticity |     $H(\mathcal{G})-H(\mathcal{G}\| \mathcal{G}\cdot \rm{M}) $    | Bernoulli distribution & Gumbel softmax reparameterization  trick |
> |      Sparsity & Regularization |                        Interdependent                     |                  Independent                  |
>
> >***Q9: …I am more interested in qualitative analysis like case studies…one last long-standing question of myself: We say explainability helps understand black-box models. But why can we trust the explanations generated by another black-box model(s)?…***
>
> Thanks for your suggestions. We have added the qualitative analysis in Appendix C. Please see if the case study clarifies your concerns.
>
> As for you concern about the black-box property of the current explainers, we agree with that it is one of the most crucial limitations of existing explainers, whereas it is surprisingly neglected by the most researches.  **It is also the original intention of our work:** though we can’t reveal the black-box property of the current explainers by leaps, we can mine the property of their components by steps. For example, to **rethink the role of regularization part** in existing explainers, instead of empirically inheriting l1-norm or l2-norm from previous works.
>
> Fortunately, there is already a few pioneering works like [Miao et al.(2022)](https://proceedings.mlr.press/v162/miao22a.html) leading the way, and I hope one day every little will make a mickle.
>
>
> Thank you again for the valuable feedback! Please let us know if you have any other comments/questions, we are more than happy to address them further!
>
> Best,
>
> Authors

---

> > ### Comment · Reviewer_HoXU · 2022-11-19
> > **Feedback**
> >
> > Thank you for the detailed response. Most of my concerns are addressed, but I am still not convinced by the sparsity one and the follow-up one on motifs.
> > - For the term $-I(G, G_s)$, why can we say it enlarges the distance between $G$ and $G_s$? I don't see any intuition from how mutual information is defined.
> > - For Q2.3, I think either the author misunderstood my question or I misunderstood some part in author response: based on my understanding, if the proposed method aims to find sparse subgraph, then it is unable to find those dense motifs as the explanation, which might be the case in some real-world scenarios (e.g., some molecular structure). Then it might further limit the applicability of the proposed method.

---

> > > ### Author Response · Authors · 2022-11-19
> > > **Response to reviewer HoXU**
> > >
> > > Dear reviewer:
> > >
> > > We especially appreciate your thoughtful comments! Below, we give responses to your comments and describe the revisions we made to address them. We are more than happy to add additional clarifications to address any additional recommendations from you!
> > >
> > > >***Q1: …why can we say it enlarges the distance between $\mathcal{G}$ and $\mathcal{G}_s$?...***
> > >
> > > Thanks for your concern. Sorry for that we don’t make it clear. Formally, mutual information is a measure of dependence or non-linear relations between two variables.
> > >
> > > Specifically, for $-I(\mathcal{G},\mathcal{G}_s)$, since $\mathcal{G}_s$ is the subgraph of $\mathcal{G}$, all edges in $\mathcal{G}_s$ also appear in $\mathcal{G}$. That is, even if we arbitrarily delete an edge in $\mathcal{G}_s$, the number of overlap edges or the correlation between $\mathcal{G}_s$ and $\mathcal{G}$ will be reduced. In this case, the target of reducing the correlation between $\mathcal{G}_s$ and $\mathcal{G}$ boils down to delete the edges in $\mathcal{G}_s$ as much as possible. It finally constrains the size of $\mathcal{G}_s$ and enlarges the distance between $\mathcal{G}_s$ and $\mathcal{G}$.
> > >
> > > >***Q2: …if the proposed method aims to find sparse subgraph, then it is unable to find those dense motifs as the explanation…***
> > >
> > > Thanks for your concern. At first, current GNNs explainability defines a subgraph $\mathcal{G}_s$ as sparse enough according to the ratio of edge numbers between $\mathcal{G}_s$ and $\mathcal{G}$. We wonder if you mistake it for the sparsity of the adjacency matrix of $\mathcal{G}_s$.
> > >
> > > In practice, even if the adjacency matrix of $\mathcal{G}_s$ only contains $1$s ($i.e.$, does not contain $0$s), it might be defined as sparse enough as long as its number of edges is much less than the number of edges in $\mathcal{G}$.
> > >
> > > Therefore, **in this way both current explanation methods and our proposed methods could find those dense motifs as the explanation.**
> > >
> > > Thank you again for the valuable feedback! Please let us know if you have any other comments/questions!
> > >
> > > Best,
> > >
> > > Authors

---

> ### Author Response · Authors · 2022-11-11
> **Response to reviewer HoXU.**
>
> Thank you for the insightful and thoughtful feedback!  We have carefully revised our paper by taking into account all your suggestions. The modified parts are highlighted in blue.
>
> Below, we give point-by-point responses to your comments and describe the revisions we made to address them. We are more than happy to add additional clarifications and revisions to the paper to address any additional recommendations from you at any moment.
>
> >***Q1: is it possible to better explain why the rank is better than $l_1$ norm?***
>
> Sorry for that we don’t make it clear. The ‘*rank of masks*’ means sorting according to the  $l_1$ norm of the mask, not the rank of the matrix comprising all the masks. We have modified our representation in Section 1 to avoid this misunderstanding.
>
> What’s more, here is the reason we argue the $l_1$ norm in current explainers:
>
> + Feature selection sequentially fills the edge with larger importance (*i.e.*, larger mask) into the salient subgraph. The key here is not the absolute magnitude of the mask (*i.e.*, $l_1$ norm), but rather the relative magnitude between the masks. Thus, we argue that the necessity of $l_1$ norm constraint needs more theoretical support.
>
> >***Q2.1: how does Eq. (1) ensure the sparsity of subgraph?***
>
> Eq. (1) ensures the sparsity of subgraph by its second term: $-I(\mathcal{G},\mathcal{G}_s)$, which constrains the mutual information between  input graph $\mathcal{G}$ and subgraph $\mathcal{G}_s$, and enlarges the distance between them [1,2].
>
> [1] Graph Information Bottleneck for Subgraph Recognition. ICLR 2021
>
> [2] Graph Information Bottleneck. NeurIPS 2020
>
> >***Q2.2: how do we define the subgraph are sparse enough?***
>
> Typically, there are two lines to constrain the sparsity of subgraph:
> + Pre-defining a threshold (*i.e.*, $\eta$ in our paper) at the beginning of explainability. In this case, while the ratio between the numbers of edges in $\mathcal{G}_s$ and $\mathcal{G}$ is less than $\eta$, we define $\mathcal{G}_s$ as sparse enough.
>
> + More recently, [Miao et al.(2022)](https://proceedings.mlr.press/v162/miao22a.html) proposes an information constraint, which is cotrained with explainers, to take the place of the above threshold. In this case, while the mutual information $ I(\mathcal{G},\rm{Y})-I(\mathcal{G},\mathcal{G}_s)$ converges to an optimal solution, we define $\mathcal{G}_s$ as sparse enough.
>
> >***Q2.3: ...for some motif structure...is very important in determining...should we consider such motifs…?***
>
> Yes, we should. Actually, extracting such motifs is the purpose of GNNs explainability. Current explainers endeavor to delete the edges not belonging to these motifs as much as possible.
>
> >***Q3: It is better to explain what $\mathcal{G}^\ast$ is in the definition.***
>
> Thanks for your suggestion. Please see the first paragraph in Section 2, where $\mathcal{G}^\ast$ is defined as the ground-truth subgraph. We have also polished the involved representation in the reversion for easy of understanding.
>
> >***Q4: …It would be great to discuss the similarity and difference with Theorem 4.1.***
>
> Thanks for your suggestion! We have added the discussion about GIBE in Section 3.1. Below we summarize the detailed differences and relation in the following table:
>
> |    |                    |      $~~~~~~~~~~~~~~~~~~~~~~~~~~~~~~~~~~~~~~~~~~~~~~~~~~~~~~~~~~~~~~~~~~~~~~~~$GIBE  &nbsp; &nbsp;               |     Theorem 4.1 in GSAT [1]   |
> |:------------------: |:---------------------:|:---------------------------:|:------------------:|
> | | | | |
> | Relation|             Theoretical Foundation |   GIB           | GIB  |
> | |           Optimal Solution | $\mathcal{G}_{s}$  |        $\mathcal{G}_{s}$      |
> | | | | |
> | Differences |          Motivation |          Unifying explainers into a common paradigm          |    Proving generalization  |
> |  |   Perspective  |   Feature attribution and selection  |   -  |
> | | Scope |           &nbsp;  &nbsp;&nbsp;  &nbsp;&nbsp;  &nbsp;&nbsp;  &nbsp;    &nbsp;  &nbsp; &nbsp;  &nbsp; Attention-based explainers (PGExplainer [2]);&nbsp;   &nbsp;&nbsp;  &nbsp;&nbsp; Mask-based explainers (GNNExplainer [3]);&nbsp;   &nbsp;&nbsp;  &nbsp;&nbsp;Perturbation-based explainers (Gem [4]);                |         GSAT [1]      |
> |  |  |  |  |
>
>
>  [1] Interpretable and Generalizable Graph Learning via Stochastic Attention Mechanism. ICML2022
>
> [2] Parameterized Explainer for Graph Neural Network. NIPS2020
>
> [3] GNNExplainer: Generating Explanations for Graph Neural Networks. NIPS2019
>
> [4] Generative Causal Explanations for Graph Neural Networks. ICML2021

---

### Author Response · Authors · 2022-11-11
**Response to all reviewers.**

Dear reviewers:

Thanks for your thoughtful and constructive review! We were encouraged to hear that our work has the important topic (Reviewer eWP9) which unifies the current explanation methods (Reviewer HoXU, SjeN) and puts these works into a novel perspective (Reviewer SjeN).  Following all suggestions for the representation (Reviewer HoXU, SjeN and eWP9), we have modified the manuscript to address all the concerns in the updated version. We would especially appreciate it if you could take another look!

Thank you once again for your valuable feedback!

Best,

Authors

---

### Comment · Area_Chair_1f9F · 2022-11-16
**Please read and respond to the rebuttals**

Dear reviewers,

The authors have responded to your reviews. Since they cannot update the submission after Nov 18, please make an effort to engage with them as soon as possible. Thank you very much!

AC

---

### Decision · Program_Chairs · 2023-01-20

**Decision:**

Reject

**Justification For Why Not Higher Score:**

One reviewer provided compelling reasons why the paper lacks clarity in the more formal parts.

**Justification For Why Not Lower Score:**

N/A

**Metareview: Summary, Strengths And Weaknesses:**

The paper studies the role of regularization in explaining GNNs through the lens of information theory.  The reviewers agree that the paper addresses an important topic by integrating and shedding light on current explanation methods. However, one reviewer made some compelling arguments about the paper's lacking rigor in several places. For instance, the reviewer identified ambiguities in the definition of certain mathematical formalisms in the paper. The reviewer, after being encouraged to be more specific, provided in my opinion strong evidence that the paper is not ready for publication. Given that the other reviewers gave borderline scores, my recommendation is that this paper should be rejected. The authors should, however, feel encouraged to continue this line of work and improve the paper.